# High fat diet ameliorates mitochondrial cardiomyopathy in CHCHD10 mutant mice

Nneka Southwell[1,2], Onorina Manzo[3], Sandra Bacman[4], Dazhi Zhao[1], Nicole M Sayles[1,2], Jalia Dash[1], Keigo Fujita[5], Marilena D'Aurelio [ID][1], Annarita Di Lorenzo [ID][3], Giovanni Manfredi [ID][1] & Hibiki Kawamata [ID][1✉]

## Abstract

Mutations in *CHCHD10*, a mitochondrial protein with undefined functions, are associated with autosomal dominant mitochondrial diseases. *Chchd10* knock-in mice harboring a heterozygous S55L mutation (equivalent to human pathogenic S59L) develop a fatal mitochondrial cardiomyopathy caused by CHCHD10 aggregation and proteotoxic mitochondrial integrated stress response (mtISR). In mutant hearts, mtISR is accompanied by a metabolic rewiring characterized by increased reliance on glycolysis rather than fatty acid oxidation. To counteract this metabolic rewiring, heterozygous S55L mice were subjected to chronic high-fat diet (HFD) to decrease insulin sensitivity and glucose uptake and enhance fatty acid utilization in the heart. HFD ameliorated the ventricular dysfunction of mutant hearts and significantly extended the survival of mutant female mice affected by severe pregnancy-induced cardiomyopathy. Gene expression profiles confirmed that HFD increased fatty acid utilization and ameliorated cardiomyopathy markers. Importantly, HFD also decreased accumulation of aggregated CHCHD10 in the S55L heart, suggesting activation of quality control mechanisms. Overall, our findings indicate that metabolic therapy can be effective in mitochondrial cardiomyopathies associated with proteotoxic stress.

**Keywords** CHCHD10; Mitochondrial Cardiomyopathy; High-Fat Diet; Mitophagy
**Subject Categories** Cardiovascular System; Genetics, Gene Therapy & Genetic Disease; Organelles

See also: H-P Lin & DP Narendra

## Introduction

Mitochondria are crucial to life and their impairment contributes to the pathogenesis of many genetic diseases associated with mutations in nuclear or mitochondrial DNA encoded genes. Pathogenic variants in the mitochondrial protein coiled-coil-helix-coiled-coil-helix domain-containing protein 10 (CHCHD10) were first associated with autosomal dominant familial amyotrophic lateral sclerosis and frontotemporal dementia (ALS-FTD) (Bannwarth et al, 2014; Johnson et al, 2014; Muller et al, 2014), and later with other diseases, including Charcot Marie Tooth neuropathy (Auranen et al, 2015), spinal muscular atrophy (Penttila et al, 2015), mitochondrial myopathy, and cardiomyopathy (Shammas et al, 2022). The normal function of CHCHD10 is still under investigation, but studies have shown that it associates with its paralog protein CHCHD2 and several other mitochondrial proteins, such as P32 (Burstein et al, 2018), mitofilin (Genin et al, 2018), and prohibitin in association with SLP2 (Genin et al, 2022). Individual ablation of *Chchd10* (Anderson et al, 2019; Liu et al, 2020) or *Chchd2* (Nguyen et al, 2021) does not impair mitochondrial function in mice, while knock out of both proteins causes mitochondrial integrated stress response (mtISR) activation and mild mitochondrial cardiomyopathy (Liu et al, 2020). This indicates that the two paralog proteins can complement each other functionally and that they are not essential for life in mammals. Together with the autosomal dominant nature of CHCHD10 mutations, these findings suggest that CHCHD10 diseases are caused by a pathogenic gain of toxic function of the mutant protein (Anderson et al, 2019). Heterozygote S55L (S59L in human) knock-in mice develop a fatal mitochondrial cardiomyopathy (Anderson et al, 2019; Genin et al, 2019; Liu et al, 2020). Recently, another knock-in mouse model harboring a G58R amino acid substitution was also shown to develop a fatal myopathy and cardiomyopathy phenotype (Shammas et al, 2022). Large CHCHD10 protein aggregates have been described in these animal models and were shown to result in activation of the mtISR accompanied by extensive metabolic rewiring (Anderson et al, 2019; Shammas et al, 2022).

During early postnatal heart development, cardiomyocyte energy metabolism shifts from predominant glucose utilization to fatty acid oxidation (FAO). However, in pathological conditions

[1]Feil Family Brain and Mind Research Institute, Weill Cornell Medicine, 407 East 61st Street, New York, NY 10065, USA. [2]Neuroscience Graduate Program, Weill Cornell Graduate School of Medical Sciences, 1300 York Ave, New York, NY 10065, USA. [3]Department of Pathology and Laboratory Medicine, Weill Cornell Medicine, 1300 York Avenue, New York, NY 10065, USA. [4]Department of Neurology, University of Miami, 1600 NW 10th Ave, Miami, FL 33161, USA. [5]Millburn High School, 462 Millburn Ave, Millburn, NJ 07041, USA. ✉E-mail: hik2004@med.cornell.edu

leading to cardiac hypertrophy, there is an uncoupling of glucose uptake and oxidation because, while the diseased heart increases glucose uptake and decreases FAO, glucose oxidation is unchanged or decreased (Ritterhoff and Tian, 2017). We found that the S55L mouse heart displays a metabolic signature, driven by mtISR, which is consistent with that described in cardiac hypertrophy conditions (Sayles et al, 2022). In the S55L heart, uncoupling of glucose uptake and oxidation accompanied by decreased FAO may have detrimental effects associated with imbalance of key metabolites. These metabolic changes could result from alterations in mitochondrial proteostasis, since they were also described in a mouse model of cardiomyopathy caused by cardiac-specific ablation of the mitochondrial protease Yme1L (Wai et al, 2015). Interestingly, in the Yme1L cardiac knockout (KO) mouse, high-fat diet (HFD) decreased insulin sensitivity, forced the utilization of fatty acids (FA), and attenuated cardiac dysfunction (Wai et al, 2015). Hence, we hypothesized that modulation of metabolic substrate preference could be beneficial for CHCHD10 cardiomyopathy. Furthermore, we anticipated that HFD would enhance the clearance of CHCHD10 aggregates by activating mitochondrial quality control mechanisms (Tan et al, 2021), thereby decreasing proteotoxic mitochondrial stress.

To test these hypotheses, we subjected heterozygote S55L mice (Het) to HFD. We found that chronic HFD ameliorated cardiac function in Het mice and significantly prolonged the lifespan of female Het mice under pregnancy-induced hemodynamic stress. Accordingly, HFD modified the metabolism of Het mice, by decreasing carbohydrates and increasing FA utilization. Importantly, in S55L mice heart oxidative phosphorylation is preserved until late in the disease course (Anderson et al, 2019; Genin et al, 2019; Liu et al, 2020; Sayles et al, 2022), which allows for reactivation of FAO by HFD. Furthermore, HFD attenuated the burden of CHCHD10 aggregates and the mtISR by promoting the autophagic clearance of damaged heart mitochondria. Taken together, our results indicate that HFD, which in wildtype (WT) mice induces mitochondrial dysfunction and metabolic syndrome (Kim et al, 2008), can be beneficial for mitochondrial cardiomyopathies.

# Results

## CHCHD10 Het mice show impaired cardiac function that is improved by HFD

To test if a HFD was cardioprotective in symptomatic Het mice, we treated male Het and WT littermate controls with a diet containing 60% fat, 20% protein, 20% carbohydrates or an isocaloric control diet (CD), starting at 190 days of age. Echocardiographic analyses were then performed two months later, at 250 days of age. Echocardiograms revealed a significant impairment of cardiac function in Het mice on CD, as shown by the reduction in fractional shortening (FS) compared to age matched WT littermates on CD (Fig. 1A,D). This result was in line with previously reported histological assessments of *Chchd10* mutant mouse heart showing evidence of cardiomyopathic changes and cardiac fibrosis (Anderson et al, 2019). Interestingly, whereas HFD did not affect FS in WT mice (Fig. 1B,D), HFD significantly improved cardiac function in Het mice (Fig. 1C,D). Other

echocardiographic measurements (LV internal diameter end diastole, LV internal diameter end systole, interventricular septum wall, and posterior wall) were unchanged in Het mice relative to WT and unaffected by the HFD in both genotypes (Appendix Fig. S1A–L). These findings indicate that HFD effectively ameliorates cardiac dysfunction in adult symptomatic animals.

Next, we wanted to determine if HFD treatment has an impact on the survival of Het mice. To this end, we took advantage of the accelerated mortality associated with cardiac failure in Het female mice undergoing pregnancy (Anderson et al, 2019). Female Het mice at 65 days of age were placed in cages with WT male mice. HFD started concomitantly with breeding and continued throughout the lifespan. Het female mice on CD and HFD had similar litter sizes (Fig. 1E) and the survival of pups to weaning was not affected by diet, except for the single mouse on CD after its fourth pregnancy (Fig. 1F). Strikingly, Het females on HFD, despite going through a higher number of pregnancies (average 3.75 vs. 2.75), survived much longer than those on CD (406.1 ± 36.9 days and 230.5 ± 49.5 days, respectively, Fig. 1G,H). Moreover, Het female mice undergoing breeding on HFD lived longer than the average lifespan of naïve females on a normal chow that we had reported previously (Anderson et al, 2019).

We also investigated if the HFD could provide protection in an independent mouse model of mitochondrial cardiomyopathy, using a conditional heart frataxin (*Fxn*) exon-2 KO mouse (*Fxn*<sup>flox/null</sup>::MCK-Cre). This mouse develops a rapidly progressing cardiomyopathy with weight loss and death at approximately 80 days of age, like the cardiac *Fxn* exon-4 KO mouse (Puccio et al, 2001). Male and female mice were treated with HFD or CD starting at 30 days of age. Surprisingly, HFD caused a small but significant decrease of body weight in both sexes (Fig. EV1A). HFD did not modify the lifespan of male mice and only modestly but significantly increased the survival of females (HFD 91.5 ± 3.7 days vs. CD 81.5 ± 3.7 days, Fig. EV1B,C). This suggests that the cardioprotective effect of HFD is context-dependent and may not extend equally to all forms of mitochondrial cardiomyopathy.

## Metabolic phenotyping studies underscore different responses to HFD in Het and WT mice

To evaluate the metabolic effects of chronic HFD we studied Het and WT mice on HFD or CD starting in utero and continued until mice reached one year of age. As previously reported (Anderson et al, 2019), the body weight of both female and male Het on CD mice ceased to increase beyond 200 days of age, whereas the body weight of WT mice continued to increase (Fig. 2A). As expected, male and female WT mice on HFD were heavier than WT mice on CD and continued to gain weight throughout the duration of the study. Het mice on HFD were heavier than Het mice on CD but lighter than WT on HFD, indicating that HFD Het mice do not accumulate as much body fat as WT and maintain a body weight similar to that of WT on CD. Furthermore, to assess the body mass composition, we performed echo-MRI in mice at 250 days of age. For these studies, we used only male mice to avoid confounding effects of hormonal cycling. Total body mass was increased in both Het and WT mice on HFD compared to CD (Fig. 2B). In both groups, HFD caused an increase of fat mass and a decrease of lean mass (Fig. 2B), suggesting that Het and WT mice on HFD undergo similar changes in body mass composition. Metabolic cage studies

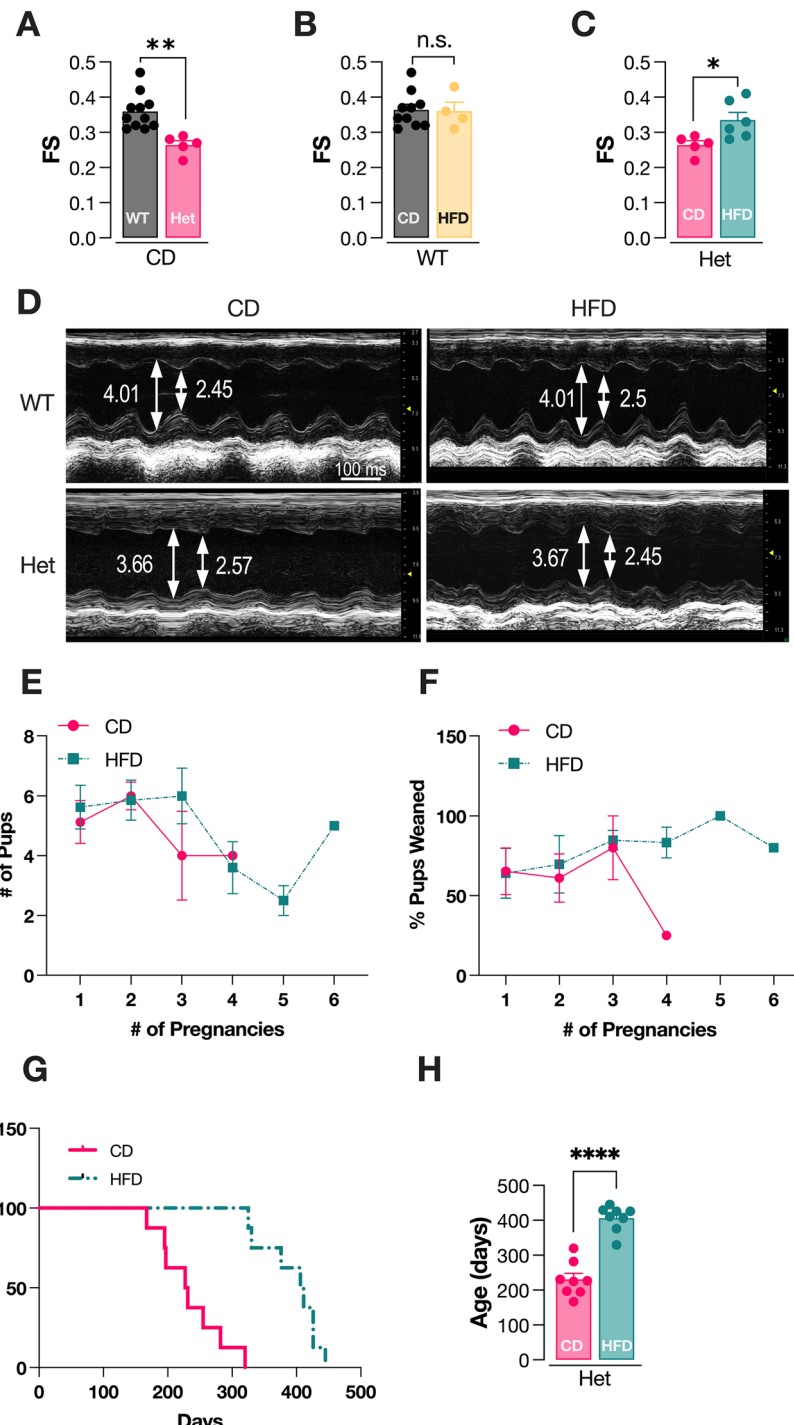

**Figure 1.  HFD attenuates cardiac dysfunction and extends lifespan in Het mice.**

(**A**) Fractional shortening (FS) measured in 250 days old WT and Het male mice on control diet (CD). (**B**) FS measured in WT male mice after 2-months of CD or HFD. (**C**) FS measured in Het male mice after 2-months of CD or HFD. (**D**) Representative images of left ventricle (LV) serial echocardiography, showing LV internal diameter end diastole (LVDd, mm) and LV internal diameter end systole (LVDs, mm) in all groups. (**E**) Average number of pups born per litter from Het dams on CD or HFD. (**F**) Average percentage of pups reared till weaning. (**G**) Kaplan–Meir survival curve plotting probability of survival against age of Het dams on CD or HFD. (**H**) Age of death of Het dams on CD or HFD. Data information: In panels (**A**), (**B**), (**C**), and (**H**), data are presented as mean ± SEM. In panels (**A–C**), WT CD ($n = 10$ mice), Het CD ($n = 5$ mice), WT HFD ($n = 4$ mice), Het HFD ($n = 6$ mice). In panels (**E**), (**F**), (**G**), and (**H**), Het CD ($n = 8$ mice), Het HFD ($n = 8$ mice). Statistical significance was determined by unpaired t-test. *$p < 0.05$; **$p < 0.01$, ****$p < 0.0001$. Source data are available online for this figure.

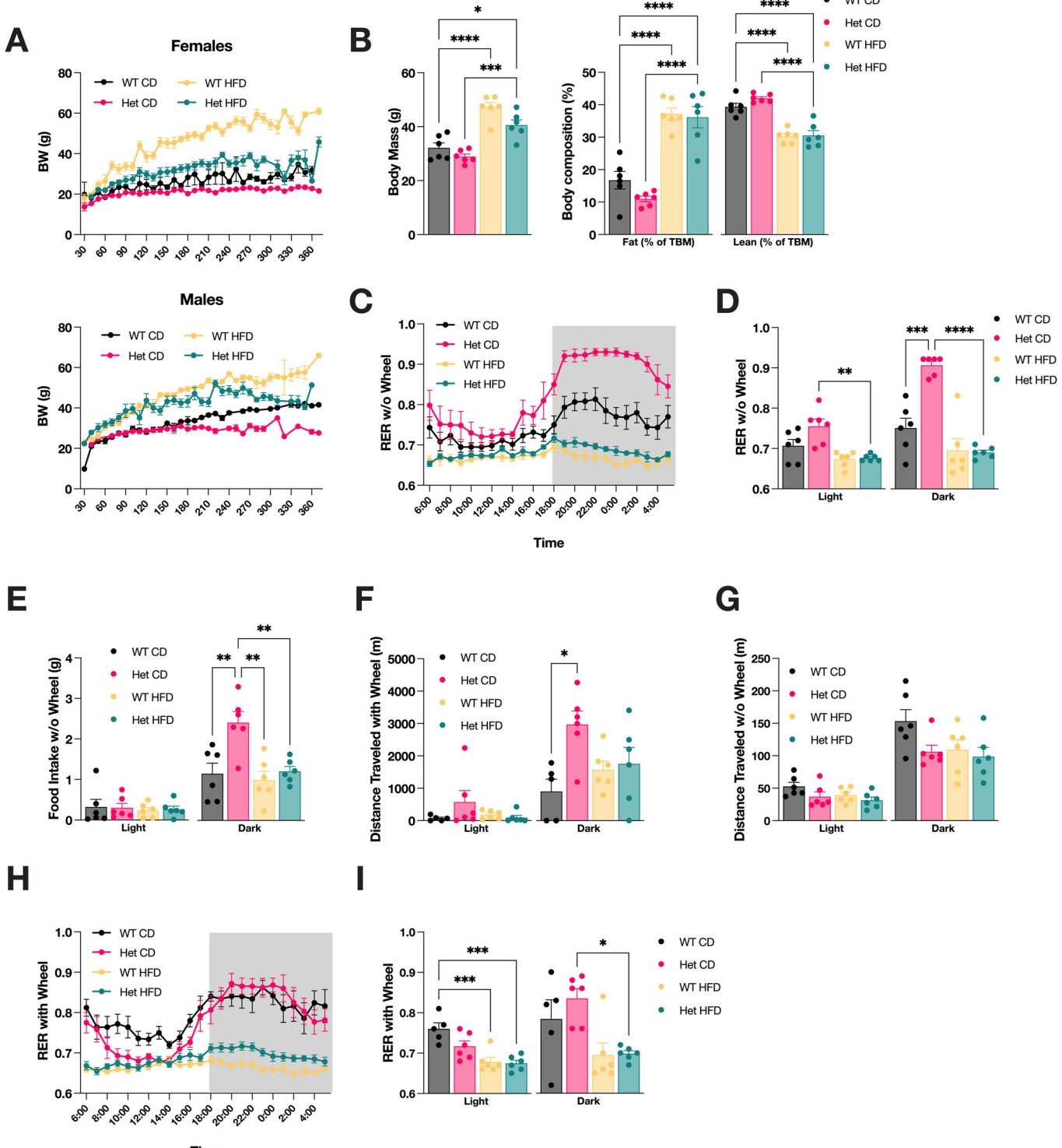

**Figure 2. The effects of HFD on metabolism differ in Het and WT mice.**

(A) Body weight changes in Het and WT mice on CD or HFD up to one year of age. (B) Body mass and composition of Het and WT male mice on CD or HFD at P250, as determined by echo-MRI. TBM = total body mass. (C) Respiratory exchange ratio (RER) of Het and WT mice measured in the Promethion metabolic phenotyping cage over 24 h (no wheel) and calculated as $VCO_2/VO_2$. (D) Average RER quantified in the light and dark cycles in cages without wheel. (E) Food intake during the light and dark cycles in cages without wheel. (F) Distance traveled on the wheel. (G) Distance traveled in cages without wheel. (H) RER over 24 h in cages with wheels. (I) Average RER in light and dark cycles, in cages with wheel. Data information: In (A), female mice, WT CD ($n = 10$), Het CD ($n = 12$), WT HFD ($n = 13$), Het HFD ($n = 12$). In (A), male mice, WT CD ($n = 8$), Het CD ($n = 8$), WT HFD ($n = 13$), Het HFD ($n = 12$). In (B–I), WT CD ($n = 6$ males), Het CD ($n = 6$ males), WT HFD ($n = 6$ males), Het HFD ($n = 6$ males). For (B), (D), (E), (F), (G), and (I), statistical significance was determined by one-way ANOVA with Tukey's correction for multiple comparisons. *$p < 0.05$, **$p < 0.01$, ***$p < 0.0005$, ****$p < 0.0001$. Data are expressed as mean ± SEM. Source data are available online for this figure.

were also performed in male mice at 250 days of age. Although the total energy expenditure in light and dark cycles was unchanged by genotype or diet (Appendix Fig. S2A), the respiratory exchange ratio (RER, i.e., the ratio between $VCO_2$ and $VO_2$) showed significantly higher values in Het mice compared to WT mice on CD (Fig. 2C,D). This suggests that a mixture of substrates including carbohydrates and proteins is utilized by Het mice on CD, while WT mice on CD metabolize predominantly fat (RER = 0.7). RER of Het mice on HFD was significantly lower than mice on CD, both in the light and in the dark, suggesting increased utilization of FAO for energy metabolism. Interestingly, in the dark, when mice are awake and consume food, RER increased in both genotypes on CD, but significantly more in the Het mice (Fig. 2C,D). Conversely, mice on HFD did not show changes in RER in the dark indicating absence of metabolic flexibility. Food intake in the light was not affected by genotype or diet (Fig. 2E), but significantly increased in Het mice on CD in the dark, an increase that was completely abolished in HFD.

Notably, when a running wheel was placed in the cage to allow for voluntary exercise, in the dark Het mice on CD ran on the wheel significantly more than WT (Fig. 2F). Instead, the distance traveled outside of the wheel in the dark was unchanged in WT and Het mice on CD (Appendix Fig. S2B). Without a wheel in the cage, the Het CD mice traveled less than WT CD mice in the dark (Fig. 2G). Het mice on CD display increased voluntary running on the wheel, which is eliminated by HFD. Traveled distance in the open field test did not differ significantly between WT and Het mice (Appendix Fig. S2C). Taken together, these results suggest that the increased running on the wheel is not part of a hyperactive phenotype of the Het mice on CD, rather it indicates a selective preference for the wheel. We then examined the RER of mice allowed to run on the wheel. In the dark, the wheel increased the RER of WT mice on CD (Fig. 2H,I), indicating enhanced utilization of carbohydrates, while the RER of Het mice remained as high as without the wheel, presumably because the carbohydrate utilization was already maximized. These data indicate that mice on HFD remain metabolically inflexible, even in the presence of the wheel.

HFD is known to alter insulin sensitivity and decrease tissue glucose uptake. We measured glucose levels and insulin sensitivity by insulin tolerance test. While fasting glucose did not differ in WT and Het mice on CD, baseline glucose levels were lower in Het mice as compared to WT on HFD (Fig. 3A). The insulin tolerance test showed that blood glucose decreased after insulin injection in WT mice on CD and the HFD induced insensitivity, as expected. However, insulin sensitivity was already decreased in Het mice on CD and worsened by HFD (Fig. 3B).

Next, to assess if HFD caused alterations of liver function consistent with excess fat accumulation (steatosis) we first analyzed liver enzymes in blood of WT and Het mice at 250 days of age. Alkaline phosphatase (ALP), aspartate transaminase (AST), and alanine transaminase (ALT) were unaffected by genotype and diet (Appendix Fig. S3A). Interestingly, the albumin/globulin ratio was decreased (Fig. 3C) and lactate dehydrogenase (LHD) increased (Fig. 3D) by HFD in WT, but not in Het mice. Other blood chemistry parameters, including bilirubin, total protein, cholesterol, and triglycerides, were unchanged in both genotypes (Appendix Fig. S3B–D). We then studied mouse liver histology at 300 days of age. Liver hematoxylin-eosin staining highlighted vacuoles indicative of fat accumulation in WT HFD and WT CD mice, and lipid accumulation was significantly less abundant

in Het mice on both CD and HFD (Fig. 3E,F). Together, these results suggest that WT mice develop liver dysfunction, whereas livers of Het mice are largely spared, likely because they do not accumulate as much fat as WT.

To investigate the systemic effects of HFD in WT and Het mice and identify potential circulating biomarkers, we performed serum metabolomic and lipidomic analyses at 250 days of age. In CD, there were only a small number of significantly different serum metabolites between WT and Het mice (Fig. EV2A), which did not define any apparent pathway. Surprisingly, HFD modified the levels of fewer metabolites in Het (Fig. EV2B,D) than in WT mice (Fig. EV2C,D). Furthermore, differentially abundant serum metabolites in HFD were largely non-overlapping in WT and Het mice (Fig. EV2D). Among the metabolites altered in Het mice, 9-Hydroxyoctadecadienoic acid (9-HODE) showed the highest statistical significance and fold change (Fig. EV2B). 9-HODE is an oxidation product of linoleic acid, which is increased by oxidative stress (Vangaveti et al, 2010). This metabolite was not significantly different between Het and WT mice in CD, suggesting that it could serve a potential biomarker to monitor the effect of HFD in Het mice. Serum lipidomic analyses showed several species of lipids that were differentially represented based on genotype and diet (Fig. EV2E). The effects of HFD on serum lipids were partially overlapping between Het and WT mice (Fig. EV2F), but there were several lipids, including phospholipids, sphingomyelins, and triglycerides that responded differentially in the two genotypes. Free fatty acids were also affected by HFD in both genotypes, as expected, but there were more significantly increased fatty acids in the serum of Het than WT mice (Fig. EV2G).

## HFD has different transcriptional and metabolic effects in WT and Het hearts

Since the mitochondrial cardiomyopathy in Het mice is associated with extensive metabolic rewiring, we reasoned that modifications of transcriptional profiles and metabolite pools could underlie the beneficial effects of HFD. To avoid the potentially confounding effects of cardiac degeneration in late disease stages, we chose to investigate mice treated with HFD throughout life in utero by feeding pregnant dams with HFD and continuing the diet after weaning. We performed unbiased transcriptomic and metabolomics analyses at 75 days of age, a time when Het hearts express cardiac stress markers (Sayles et al, 2022), but no symptoms of cardiac failure.

As expected, many genes were significantly different between Het and WT hearts on CD (4621 DEGs with $p_{adj.} < 0.05$, Fig. 4A). Remarkably, HFD had much larger transcriptional effects in Het hearts (4838 DEGs, Fig. 4B) than in WT hearts (33 DEGs, Fig. 4C), suggesting that mutant hearts respond to HFD differently than WT hearts. Compared to Het mice on CD vs. WT CD (Fig. 4A), there were less DEGs when Het mice were in HFD (1082 DEGs, Fig. 4D), suggesting that HFD normalizes expression of genes altered in mutant hearts. As expected, we found cardiac and muscle hypertrophy in response to stress among the upregulated pathways enriched in Het vs. WT hearts on CD (Fig. 4E). Importantly, in Het mice on HFD these pathways were downregulated compared to Het mice on CD (Fig. 4F), indicating that HFD decreases cardiac stress in Het mice. Accordingly, individual genes comprising this pathway, including *Myh6*, *Nppb*, and *Atp2b4*, were decreased by HFD in Het hearts (Fig. 4G). The downregulation of two of the key cardiomyopathy stress genes, *Myh6* and *Atp2b4* (PMCA), in Het

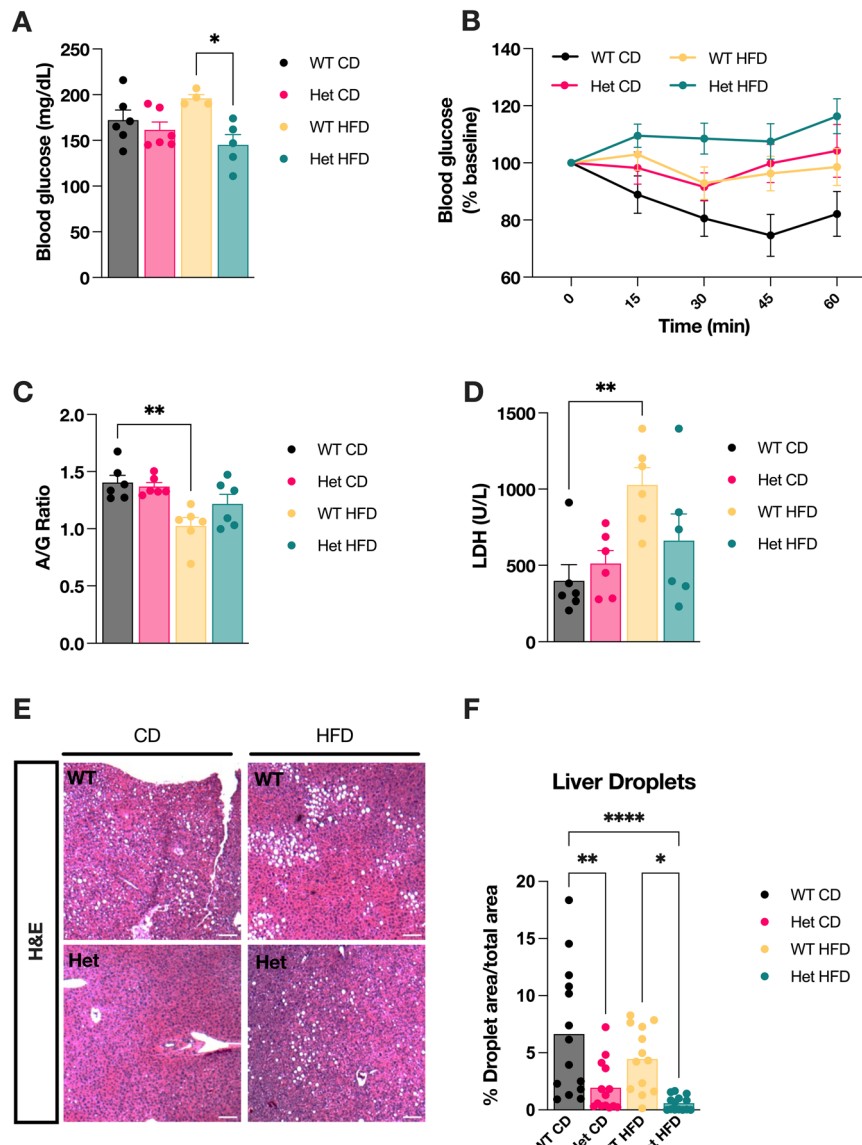

**Figure 3. HFD induces insulin resistance in both Het and WT mice, but with different systemic effects.**

(A) Glucose levels in Het and WT male mice on CD or HFD at P250, upon fasting for 18 h. (B) Upon addition of a bolus of insulin (time 0), clearance of blood glucose was monitored for 60 min. (C) Serum albumin to globulin ratio in female Het and WT mice on CD or HFD. (D) Serum lactate dehydrogenase levels (LDH) in female Het and WT mice on CD or HFD. (E) Hematoxylin and eosin staining of liver at 4× magnification at P250; scale bar, 100 μm. (F) Quantification of liver droplet area relative to total tissue area. Data information: In (A–D), WT CD ($n = 6$ mice), Het CD ($n = 6$ mice), WT HFD ($n = 6$ mice), Het HFD ($n = 6$ mice). In (F), WT CD ($n = 2$ mice), Het CD ($n = 2$ mice), WT HFD ($n = 2$ mice), Het HFD ($n = 2$ mice); quantification was performed on 14 image fields from 2 mice per group. For (A), (C), (D), and (F), statistical significance was determined by one-way ANOVA with Tukey's correction for multiple comparisons. *$p < 0.05$, **$p < 0.01$, ***$p < 0.0005$, ****$p < 0.0001$. Data are expressed as mean ± SEM. Source data are available online for this figure.

mice treated with HFD was confirmed at the protein level (Fig. EV3A–D).

There were also significant differences in metabolite abundance between Het and WT hearts on CD (64 metabolites with Log2 fold change >1; $p < 0.05$, Fig. 5A). Whereas HFD induced similar numbers of metabolic changes in Het (142 metabolites with $p < 0.05$, Fig. 5B) and WT (147 metabolites with $p < 0.05$, Fig. 5C) hearts, most of the metabolites did not overlap (Fig. 5D), again suggesting differential effects of HFD on Het and WT hearts. Enrichment analysis of all significant ($p < 0.05$), non-overlapping metabolites in HFD showed

notable pathways enriched only in WT hearts, including glycolysis, citric acid cycle, and carnitine biosynthesis (Fig. 5E,F). Pathways enriched only in Het on HFD, included upregulation of phosphatidylcholine and phosphatidylethanolamine biosynthesis and downregulation of cardiolipin biosynthesis (Fig. 5G,H). A heatmap of the 50 most differentially represented metabolites highlighted groups of molecules which differed based on genotype and others that differed based on diet, and a group that depended on both genotype and diet (Fig. 5I). Interestingly, most of these metabolites responded differently to HFD in WT and Het hearts. Among these, a subset was changed by

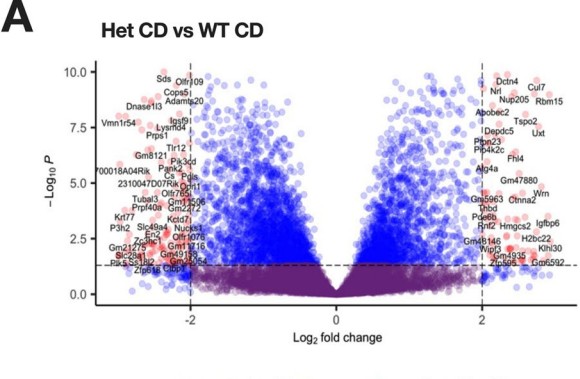

A **Het CD vs WT CD**

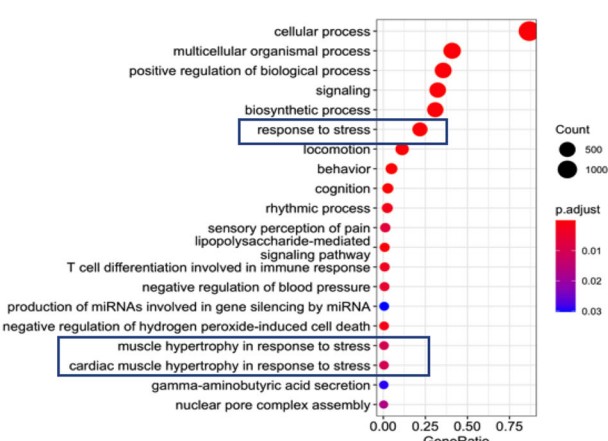

E **Upregulated Pathways in Het CD vs WT CD**

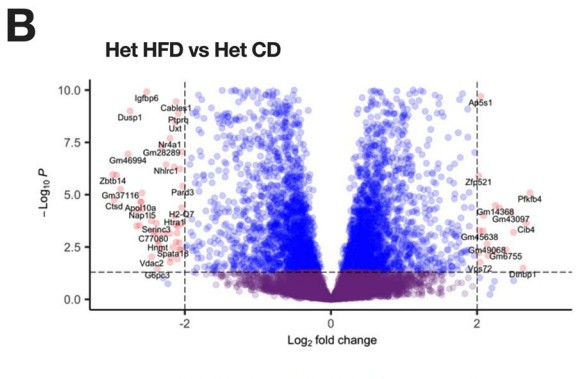

B **Het HFD vs Het CD**

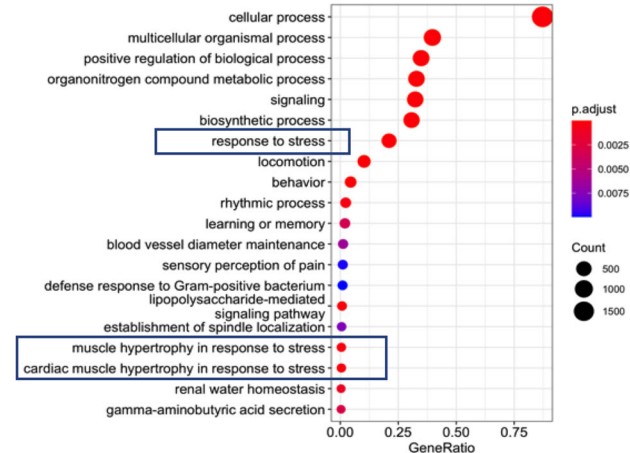

F **Downregulated Pathways in Het HFD vs Het CD**

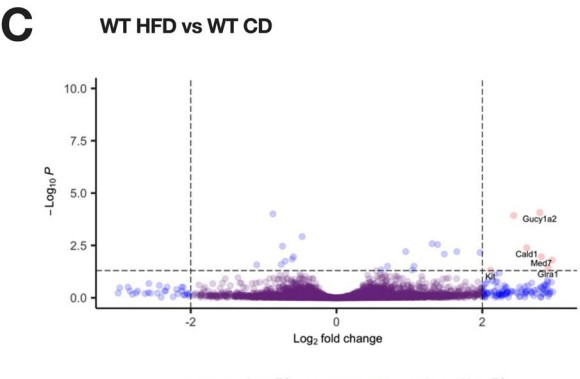

C **WT HFD vs WT CD**

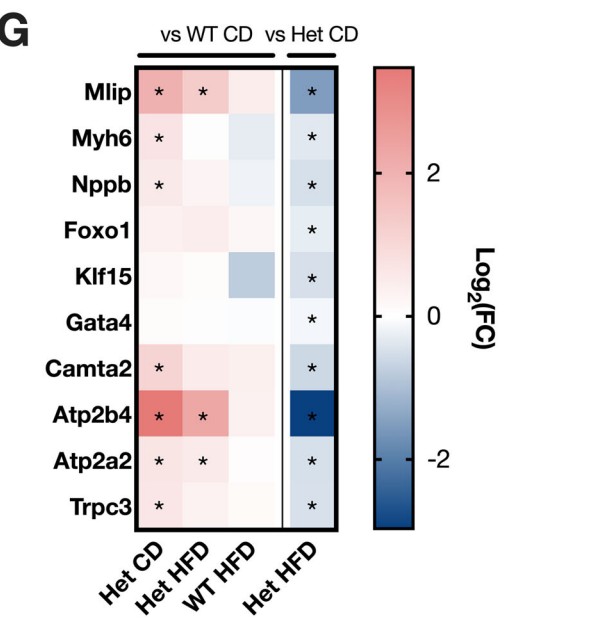

G

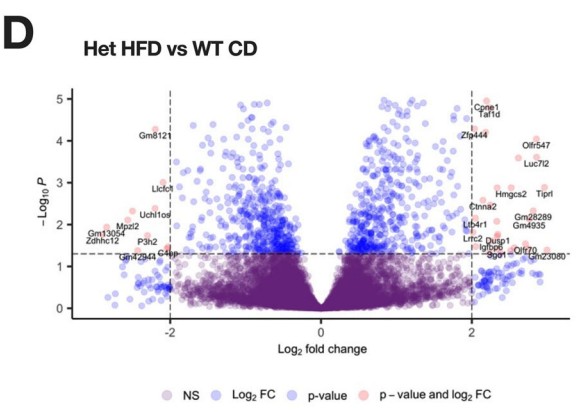

D **Het HFD vs WT CD**

◀  **Figure 4.  Transcriptomic profiles suggest that HFD attenuates cardiac stress in Het mice.**

(A) Volcano plot of heart DEGs between Het and WT mice on CD. (B) Volcano plot of heart DEGs between Het HFD vs. Het CD. (C) Volcano plot of heart DEGs between WT HFD vs. WT CD. (D) Volcano plot of heart DEGs between Het HFD vs. WT CD. (E) Dot plot of pathway enrichment of all upregulated DEGs between Het CD vs. WT CD in heart. (F) Dot plot of overrepresented pathways with downregulated DEGs between Het HFD and Het CD hearts. (G) Heatmap showing effect of HFD on genes identified in the stress induced cardiac hypertrophy pathway. Data information: In (A–G), WT CD ($n = 10$ mice), Het CD ($n = 10$ mice), WT HFD ($n = 10$ mice), Het HFD ($n = 10$ mice). Equal numbers of female and male mice were studied. For (A–D), the threshold for volcano plots were set at $log_2FC = 2$ and $p_{adj} < 0.05$. For (E) and (F), upregulated and downregulated DEGs reported were set at $p_{adj} < 0.05$. Statistical significance was determined by Wald's Test for DEGs. *$p_{adj} < 0.05$.

HFD only in Het hearts and acquired a profile more alike WT in CD (Fig. 5I, group 2). Taken together, metabolomics and transcriptomics data indicate that HFD had significant and distinct effects in Het and WT hearts and that metabolic changes induced by HFD in Het mice were associated with decreased cardiac stress markers.

To further understand the drivers of the metabolic effects of HFD, we examined the expression of key genes in FAO and glycolysis. The expression of the plasma membrane FA transporter *Fatp1* (Fig. 6A) was decreased in Het in CD relative to WT hearts and increased by HFD to levels comparable to WT (Fig. 6B). The expression of *Cpt1* and *Cpt2*, which transport FA into mitochondria for oxidation (Fig. 6A), was also decreased in Het hearts and increased by HFD (Fig. 6B). The expression of *Hadha* subunit of the trifunctional protein, a crucial step of long chain FAO (Fig. 6A) and cardiolipin remodeling in the heart, was severely decreased in Het hearts and restored by HFD (Fig. 6B). Interestingly, HADHA deficiency has been associated with neonatal syndromes characterized by hypoglycemia, myopathy, and cardiomyopathy (Prasun et al, 1993). The levels of the insulin-sensitive glucose transporter *Glut4* were increased in Het hearts on CD and significantly decreased by HFD (Fig. 6C). Hexokinase isoform 1 (*Hk1*) and the pyruvate dehydrogenase subunit E1β (*Pdhb*) were decreased in Het hearts on CD and increased by HFD (Fig. 6C). For glycolytic metabolites, glucose levels were reduced in Het hearts on CD but not on HFD (Fig. 6D). Other differences in glycolytic metabolites were observed on HFD, such as higher pyruvate and lower phosphoenolpyruvate levels in Het hearts (Fig. 6D). Taken together, these findings indicate that HFD enhances FAO in Het hearts but also increases glucose oxidation.

We previously demonstrated that cardiomyopathy in Het mice is associated with upregulation of the serine one-carbon metabolism and antioxidant enzymes (Anderson et al, 2019; Sayles et al, 2022). We confirmed these findings in Het hearts on CD, showing elevation of the expression of several enzymes of this pathway, including *Psat1*, *Mthfd1l*, *Mthfd2*, *Mthfr*, and showed that HFD decreases their levels (Fig. 6E,F). Markers of mtISR and antioxidant response were also downregulated by HFD, including *Ddit3* and *Nfe2l2* (Fig. EV4A). A decrease in transcription of many respiratory chain genes, previously reported in Het hearts (Sayles et al, 2022), was mostly confirmed in this study at 75 days age, and HFD normalized the expression of several of these genes relative to WT hearts on CD (Fig. EV4B–F). Since mtDNA was found to be decreased in Het hearts (Sayles et al), we measured mtDNA levels in heart at 75 days of age. While we confirmed that Het CD mtDNA was decreased compared to WT CD, HFD did not affect mtDNA levels in either genotype (Fig. EV4G).

Het hearts on CD showed alterations of metabolites of the serine one-carbon metabolism with elevation of serine, glycine, methionine, cystathionine, GSSG, and GSH and a decrease of S-Adenosyl-L-homocysteine (SAH) (Fig. 6E,G). The levels of these metabolites were

differently modified by HFD in Het and WT hearts. For example, SAH was decreased by HFD in WT but not in Het hearts, resulting in similar levels in the two genotypes. Conversely, phosphoserine was elevated by HFD in WT but not in Het hearts. We also found that Het had decreased levels of adenosine, which were restored by HFD (Fig. 6H). This is potentially significant for the therapeutic role of HFD, since adenosine plays important roles not only in energy metabolism, but also in cardioprotection (Kitakaze and Hori, 2000). We examined several intermediates of energy metabolism and found that NADH levels were decreased in Het on CD, and interestingly HFD decreased NADH in WT but not in Het hearts (Fig. EV5A). ATP levels were unchanged in Het on CD and increased by HFD in WT by not in Het hearts (Fig. EV5B). No changes in the levels of phosphocreatine were associated with genotype or diet (Fig. EV5C). Together, these data suggest that Het hearts are not depleted of energy and HFD has different effects on the metabolism of WT and Het, whereby the levels of key metabolites were changed in WT but not Het hearts, with the exception of adenosine. Therefore, the effects of HFD on heart function and stress in Het mice are likely unrelated to energy production.

The effects of S55L D10 mutation on the lipidomic profiles of the heart had not yet been investigated. Overall, Het on HFD have more upregulated lipids compared to WT while downregulated lipids are similar amongst groups (Fig. 7A). Whereas no differences in triglycerides (Appendix Fig. S4A) and cholesterol esters (Appendix Fig. S4B) were associated with genotype or diet, we found that phospholipids were the most changed in Het on CD and in response to HFD (Fig. 7B). Many phospholipids were affected by genotype but not by diet (Fig. 7B, Group 1) while others were only affected by HFD in Het but not in WT hearts (Fig. 7B, Groups 2 and 3). Cardiolipin (CL), one of the most abundant mitochondrial inner membrane phospholipids, is of particular interest due to its role in the maintenance of mitochondria structure and function. Several CLs showed different abundance in WT and Het hearts and changed with HFD in one genotype but not the other (Fig. 7C, Groups 1–4). Transcriptomic analysis of enzymes involved in lipolysis and phospholipid biosynthesis (schematically summarized in Fig. 7D) were mostly downregulated in Het compared to WT on CD and were restored by HFD (Fig. 7E). These results show that Het hearts have altered phospholipid levels, which can contribute to mitochondrial dysfunction and cardiomyopathy, and that HFD modulates phospholipid metabolism in the heart.

## HFD decreases CHCHD10 protein levels and aggregation in Het hearts

Metabolic studies in Het mice showed that HFD induces a rearrangement in metabolism that enhances FAO, and associates with a decrease of cardiomyopathy markers. To further understand the molecular mechanism underlying the cardioprotective effects of HFD, we investigated if HFD also affects CHCHD10 at the protein

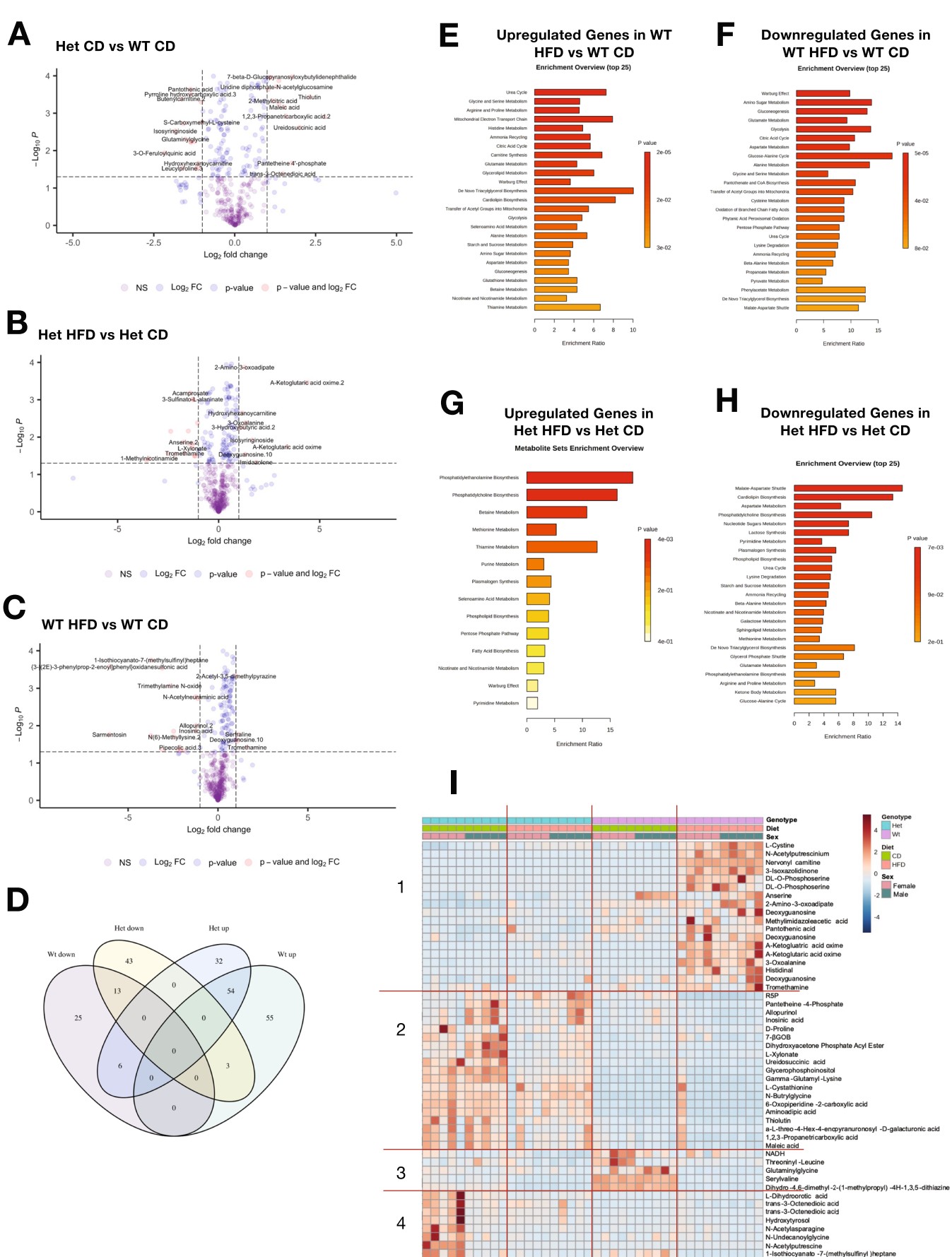

**Figure 5.  Heart metabolite profile changes induced by HFD suggest increased FA utilization in Het mice.**

(A) Volcano plot showing differential abundance of heart metabolites from Het and WT mice on CD. (B) Volcano plot with differential abundance of metabolites between Het HFD vs. Het CD. (C) Volcano plot with differential abundance of metabolites from WT HFD vs. WT CD. (D) Venn diagram of metabolites both up- and down-regulated by HFD within each genotype separately (Het HFD vs Het CD, WT HFD vs WT CD). (E, F) Pathways that were upregulated (E) and downregulated (F) by HFD only in WT heart (WT HFD vs. WT CD). (G, H) Upregulated (G) and downregulated (H) pathways only in the Het HFD vs. Het CD. (I) Heatmap of the top 50 differentially abundant metabolites in Het and WT heart on CD or HFD. R5P, Ribose-5-Phosphate; 7-βGOB, 7-beta-D-Glucopyranosyl oxy butylidenephthalide. Data information: In (A–I), WT CD ($n = 10$ mice), Het CD ($n = 10$ mice), WT HFD ($n = 10$ mice), Het HFD ($n = 10$ mice). Equal numbers of female and male mice were studied. For (A–C), thresholds were set at $log_2FC = 1$ and $p_{raw} < 0.05$. For (D–H), unpaired t-test was used within each genotype. For (I), metabolites were ranked by two-way ANOVA. The red lines and the numbers on the left denote different groups of metabolites that change based on genotype, diet, and sex.

level. Thus, we analyzed the steady-state levels of CHCHD10 in heart mitochondria. Interestingly, Western blots of mitochondrial fractions showed that HFD decreased the amount of CHCHD10 in Het hearts (Fig. 8A,B). Filter trap assays of Het HFD heart mitochondria showed a decrease in the amounts of insoluble CHCHD10 aggregates (Fig. 8C,D). Since CHCHD2 has been found to co-aggregate with CHCHD10 in Het hearts (Anderson et al, 2019), we analyzed insoluble CHCHD2 by filter trap and found that HFD also decreased CHCHD2 aggregates (Fig. 8C,D). On the other hand, quantification of *Chchd10* mRNA showed no differences between Het hearts in CD and HFD (Fig. 8E), indicating that the decrease in CHCHD10 protein was not due to transcriptional changes associated with HFD. Together, these results suggest that HFD decreases the accumulation and aggregation of CHCHD10 and CHCHD2 in Het hearts.

We hypothesized that HFD could enhance the clearance of damaged heart mitochondria containing mutant CHCHD10 aggregates through activation of mitophagy. To test this hypothesis, we examined autophagy markers in heart mitochondria. The levels of p62 (SQSTM1) associated with mitochondria were increased in Het hearts in CD, as previously reported (Anderson et al, 2019), and significantly decreased by HFD (Fig. 8F,G). The levels of LC3B-II were also increased in Het hearts in CD and decreased by HFD (Fig. 8H,I). As expected, only the lipidated form of LC3B (LC3B-II) was detected in the mitochondrial fraction, as shown by comparison with total heart lysate (Lys lane in Fig. 8H). We then examined the mitochondrial content of two key mitophagy players, the E3-ubiquitin ligase Parkin, and the ubiquitin-kinase PINK1. The levels of both Parkin (Fig. 8J,K) and PINK1 (Fig. 8L,M) were elevated in Het in CD and again decreased significantly by HFD. Of note, PINK1 was almost undetectable in all mitochondrial samples, except for Het in CD. We interpret these findings as the result of delayed clearance of mitochondria tagged for degradation in Het CD hearts, whereas degradation was enhanced by HFD, as shown by the decrease of mitophagy markers in mitochondria.

We then assessed the levels of long versus short forms of OPA1, whose ratio is known to be decreased in CHCHD10 mutant hearts (Liu et al, 2020; Shammas et al, 2022). We confirmed the increase of OPA1 cleavage in Het heart homogenates in CD, whereas in HFD the long vs. short OPA1 ratio was not different from WT (Fig. 8N,O). Furthermore, the levels of the protease OMA1, responsible for OPA1 processing, were significantly decreased in Het hearts on CD, suggesting activation and self-cleavage induced by mitochondrial stress (Shammas et al, 2022), but they were unchanged in Het in HFD relative to WT (Fig. 8P,Q).

Lastly, we examined morphological features of cardiomyocytes by electron microscopy of hearts from 300-day-old mice, which had been on CD or HFD since the age of 200 days. Electron microscopy

showed clear evidence of ongoing mitophagy of mitochondria with abnormal cristae in Het mice on HFD (Fig. 8R). Moreover, while WT cardiomyocytes on HFD accumulated large lipid droplets, the cardiomyocytes of Het mice on HFD did not show such large lipid accumulation. Overall, these data suggest that, in addition to stimulating FAO, HFD increases mitophagy in Het hearts, thereby decreasing the steady-state levels of p62, LC3B-II, PINK1, Parkin, and most importantly aggregated CHCHD10.

## Discussion

Mitochondrial proteotoxic stress causes the activation of the protease OMA1 resulting in OPA1 cleavage and remodeling of cristae structure. It also activates ISR through regulation of ATF4 translation by cleavage and cytosolic translocation of DELE1, another target of OMA1 (Fessler et al, 2020; Guo et al, 2020). ISR was recently reported as being protective, at least initially, in models of mitochondrial cardiomyopathy, including CHCHD10 mutations (Shammas et al, 2022) and cytochrome c oxidase deficiency (Ahola et al, 2022). In parallel, and potentially independent of OMA1 cleavage, mitochondrial stress also causes extensive metabolic rewiring (Kuhl et al, 2017; Shammas et al, 2022). Metabolic rewiring in these models of mitochondrial cardiomyopathy involves enhanced one-carbon metabolism and a decrease in FAO. Therefore, the substrate utilization of Het hearts is reminiscent of the energy metabolism of the fetal heart (Ritterhoff and Tian, 2017). Since HFD forces the heart to use FAO by causing insulin resistance (Wai et al, 2015), we sought to obtain proof of concept that HFD could exert a protective role in the cardiomyopathy associated with S55L mutant CHCHD10. Chronic HFD improved the cardiac function of symptomatic mutant CHCHD10 mice and life expectancy in a pregnancy-induced model of accelerated cardiomyopathy.

Importantly, the metabolic effects of HFD differed significantly between WT and mutant CHCHD10 animals. Contrary to WT mice, Het mice did not become obese even after long-term HFD, over the course of one year. Het mice did not develop liver steatosis or hepatic function alterations typically associated with HFD in WT animals. As expected, HFD caused a decrease in RER in both WT and Het mice, consistent with increased FAO and decreased carbohydrate usage. However, transcriptomics and metabolomics studies of the heart evidenced profound differences in response to HFD in WT and Het mice. Notably, pathways indicative of stress and cardiomyopathy markers were downregulated only in Het mice, further supporting the cardioprotective effects of HFD.

Taken together, the findings from this study support the cardioprotective role of HFD in mitochondrial cardiomyopathy,

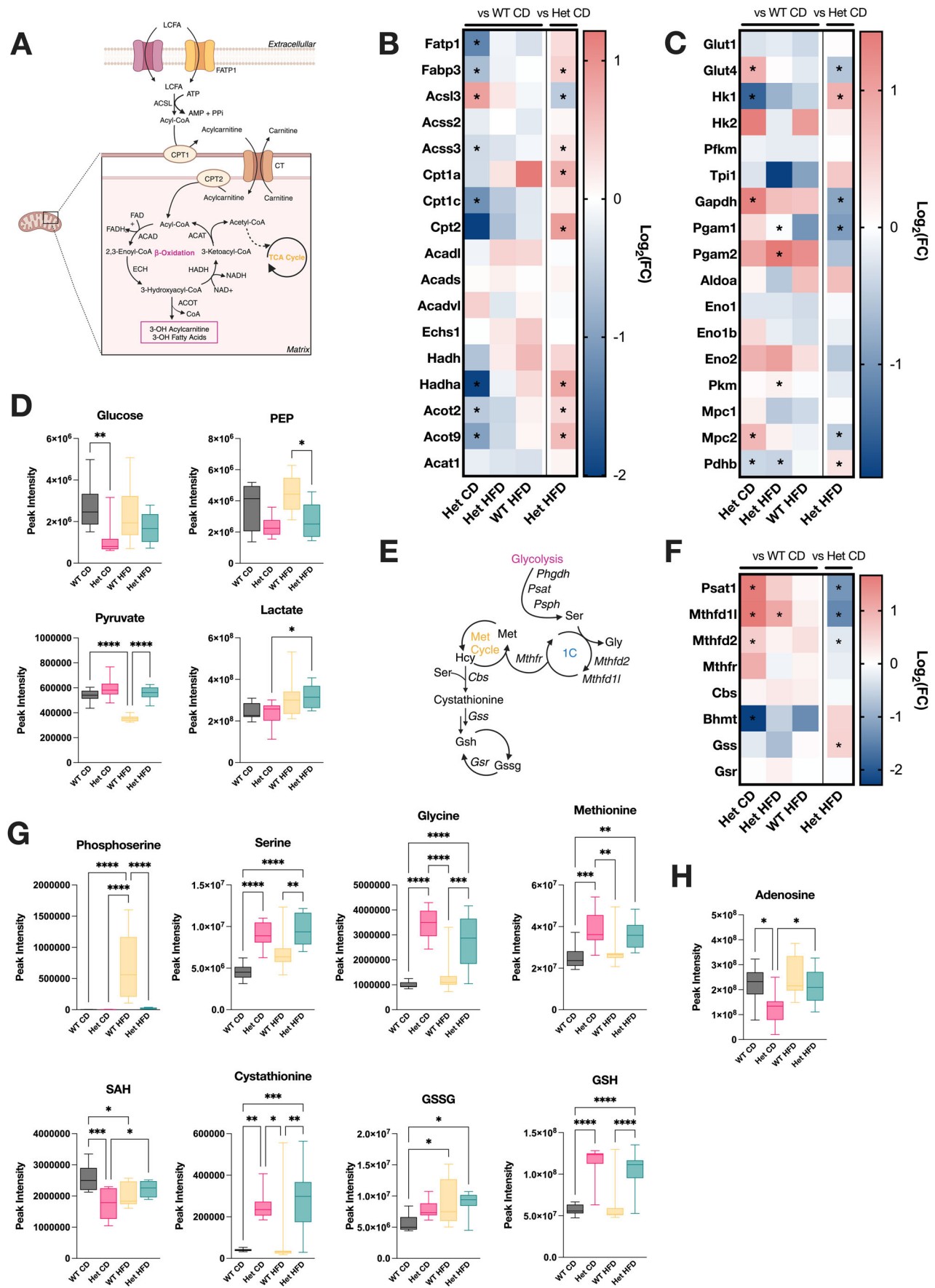

◀  **Figure 6.   HFD modulates glycolysis, FAO, and 1C-metabolism pathways in Het hearts.**

(**A**) Scheme of FAO pathway. LCFA, long chain fatty acid; FATP1, fatty acid transport protein 1; CTP, carnitine palmitoyltransferase; CT, carnitine-acylcarnitine translocase; ACAD, acyl-CoA dehydrogenase; ECH, enoyl-CoA hydratase; HADH, hydroxyacyl-CoA dehydrogenase; ACAT, acetyl-CoA acetyltransferase. (**B**) Heatmap of the expression of genes encoding enzymes of the FAO pathway. (**C**) Heatmap of genes encoding for enzymes in the glycolytic pathway. (**D**) Glycolytic metabolite levels in heart. (**E**) Scheme of one-carbon metabolism pathway. (**F**) Heatmap of genes for one-carbon metabolism enzymes. (**G**) Intermediate metabolite levels in the one-carbon metabolism pathway in the heart. (**H**) Adenosine levels in the heart. Data information: For (**B–H**), WT CD ($n = 10$ mice), Het CD ($n = 10$ mice), WT HFD ($n = 10$ mice), Het HFD ($n = 10$ mice). Equal numbers of female and male mice were studied. For (**D**), (**G**), and (**H**), statistical significance was determined by one-way ANOVA with Tukey's correction for multiple comparisons. *$p < 0.05$, **$p < 0.01$, ***$p < 0.0005$, ****$p < 0.0001$. Data are expressed as box plots showing minimum to maximum values, median, upper and lower quartiles. For (**B**), (**C**), and (**F**), asterisks denote significantly different genes ($p_{adj} < 0.05$ by Wald's test). Heatmaps were derived from transcriptomic datasets. Source data are available online for this figure.

but the benefits of HFD in mitochondrial cardiomyopathy models are likely context-dependent. Mice with conditional deletion of *Fxn* heart and skeletal muscle develop a severe fatal mitochondrial cardiomyopathy (Puccio et al, 2001), but HFD fails to extend the survival of cardiac-specific *Fxn* KO male mice and had only a small, albeit statistically significant, life extension effect in females. Although these findings indicate an interesting sexual dimorphism in the response to HFD in the *Fxn* KO mice, which deserves further investigation, they also demonstrate context-dependency for the protective effect of HFD. The reasons for the different outcomes in mutant CHCHD10 and *Fxn* KO mice likely lie on the mechanisms underpinning the mitochondrial stress. The loss of frataxin causes FeS cluster assembly defects that compromise several enzymes necessary for oxidative phosphorylation (Rotig et al, 1997). Since FAO requires a viable oxidative phosphorylation machinery, it cannot be reactivated in *Fxn* KO hearts. On the other hand, the cause of stress in mutant CHCHD10 mice is proteotoxic in nature and oxidative phosphorylation function is largely preserved until disease end stage (Anderson et al, 2019; Sayles et al, 2022), thereby allowing for reactivation of FAO by HFD.

Lipidomics studies highlighted phospholipid alterations in the Het hearts relative to WT in CD. These differences had not been described before in CHCHD10 mutant mice and are of potential disease relevance because phospholipids play major structural and signaling roles that may affect mitochondria directly (Funai et al, 2020). Especially noteworthy was the alteration of CL levels in the Het heart, since CL is a fundamental component of the mitochondrial inner membrane. Interestingly, aberrant mitochondrial cristae structure has been described in in vivo and in vitro models of pathogenic CHCHD10 mutations (Genin et al, 2022), which could be at least in part due to CL abnormalities. In HFD, a subset of phospholipids in the Het hearts become more similar in abundance to WT hearts, suggesting that the normalization of the profiles of this class of lipids may participate in the cardioprotection afforded by HFD. In serum, the differences in lipid content between Het and WT mice were not as well delineated as in heart in CD, but upon HFD treatment differences in the levels of several phospholipids as well as triglycerides became evident, suggesting that changes induced by mutant CHCHD10 in response to HFD at the systemic level could involve several tissues, including the adipose tissue, liver, and skeletal muscle. Intriguingly, in mice CHCHD10 has been implicated in mitochondrion-dependent browning of adipocytes (Xia et al, 2022) and lipolysis associated with thermogenesis (Ding et al, 2022).

We were especially intrigued to find decreased steady-state protein levels of CHCHD10 in heart mitochondria of HFD-treated Het mice as well as decreased amounts of insoluble, aggregated

CHCHD10. Since it was shown that mutant CHCHD10 impairs PINK1-Parkin-mediated mitophagy in the brain of transgenic mice (Liu et al, 2023), it is possible that a similar impairment occurs in the heart of our CHCHD10 S55L knock-in mouse. Based on our biochemical and ultrastructural findings in the Het hearts on HFD, we propose that increased turnover of damaged mitochondria could contribute to increased clearance of CHCHD10 aggregates. The decrease in OMA1 and OPA1 cleavage in Het hearts in HFD could be the result of accelerated elimination of mitochondria which have accumulated protein aggregates and become prone to stress-associated cristae remodeling. In support of this interpretation, HFD was shown to increase mitophagy in a mouse model of diabetic cardiomyopathy, where general autophagy was upregulated for up to 8 weeks of HFD and then it decreased, while an enhancement of mitophagy persisted throughout the duration of the chronic HFD treatment (Tong et al, 2019). Since RNA markers of mitochondrial biogenesis, *Ppargc1a* and *Ppargc1b* (Fig. EV4A), were elevated in Het mice on HFD, increased mitophagy could be balanced by enhanced mitochondrial biogenesis, thereby improving mitochondrial homeostasis. This hypothesis will need to be further explored in the future, by investigating mitophagy fluxes, as well as mitochondrial biogenesis, in Het and WT hearts in CD and HFD.

Overall, the positive outcome of HFD treatment in the CHCHD10 mouse model suggests a path forward toward metabolic therapy approaches for certain forms of mitochondrial diseases, but there are limitations that will need to be addressed experimentally in future work. For example, it will be necessary to investigate the survival effects of HFD in adult male Het mice. We did not perform survival studies in these mice, because we used the more aggressive pregnant female phenotype to establish proof of concept. As the disease worsens in older animals of either sex, the effects of HFD may wane because mitochondrial damage starts impairing oxidative phosphorylation (Anderson et al, 2019) and the heart can no longer use FAO efficiently. Therefore, the effects of HFD initiated at disease onset and continued until death will need further investigation in both sexes.

The finding of cardioprotection by HFD in a mouse model of mitochondrial cardiomyopathy associated with protein aggregation and mtISR suggests potential therapeutic avenues to be explored in human patients. However, the translatability of these findings to humans affected by CHCHD10 remains to be determined. Potentially, a HFD strategy could be employed in the pedigrees with CHCHD10 mutations that present with cardiomyopathy (Shammas et al, 2022), whereas it is uncertain whether HFD could have beneficial effects in patients with predominantly neurological manifestations. On the other hand, HFD could be considered as a

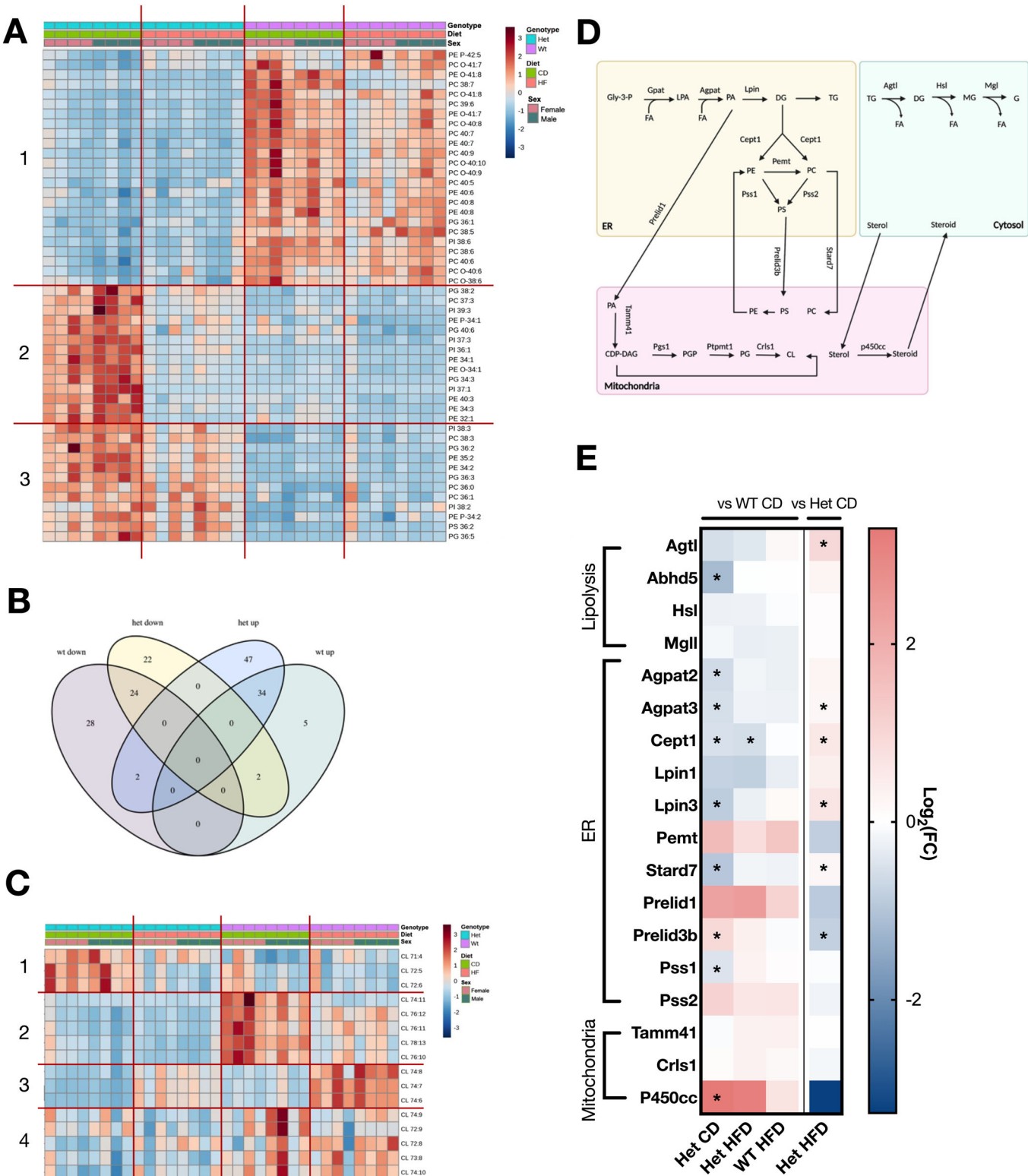

therapeutic approach in other forms of mitochondrial cardiomyopathy caused by similar pathogenic mechanisms, as long as the oxidative phosphorylation machinery is preserved.

HFD has traditionally been considered a modeling tool to elicit metabolic syndrome and its pathological consequences in experimental animals. However, we propose that in the case of CHCHD10 cardiomyopathy, and possibly other similar disorders, HFD has fewer detrimental effects, because of the underlying metabolic conditions. This notion is supported by the observation that HFD does not cause severe obesity and hepatic alterations in

**Figure 7. HFD has different effects on lipid profiles in Het and WT hearts.**

(A) Heatmap of top 50 phospholipids in Het and WT heart on CD or HFD. (B) Venn diagram of lipids that were up- and down-regulated (p < 0.05) by HFD within each genotype, separately (Het HFD vs Het CD, WT HFD vs WT CD). (C) Heatmap of cardiolipins in Het and WT heart on CD or HFD. (D) Scheme of phospholipid biosynthesis. Gly-3-P, glycerol-3-phosphate; LPA, lysophosphatidic acid; PA, phosphatidic acid; DG, diacylglycerol; TG, triacylglycerol; MG, monoacylglycerol; G, glycerol; PE, Phosphatidylethanolamine; PC, Phosphatidylcholine; PE, Phosphatidylethanolamine; PS, Phosphatidylserine; PGP, Phosphatidylglycerol phosphate; CDP-DAG, Cytidine diphosphate diacylglycerol; PG, Phosphatidylglycerol; CL, cardiolipin; FA, fatty acids. (E) Heatmap of genes involved in phospholipid biosynthesis. Data information: In (A–E), WT CD (n = 8 mice), Het CD (n = 8 mice), WT HFD (n = 8 mice), Het HFD (n = 8 mice). Equal numbers of female and male mice were studied. For (A) and (C), two-way ANOVA ranked the top 50 phospholipids. The red lines and the numbers on the left denote different groups of metabolites that change based on genotype, diet, and sex. For (B), data. For (E), asterisks denote significantly different genes ($p_{adj}$ < 0.05) by Wald's Test.

Het mice. Nevertheless, alternative pharmacological strategies to enhance FAO and decrease glucose utilization in mitochondrial cardiomyopathies could be developed in the future and potentially used in conjunction with other therapeutic approaches.

# Methods

## Animals

All animal procedures were conducted in accordance with Weill Cornell Medicine (WCM) Animal Care and Use Committee and performed according to the Guidelines for the Care and Use of Laboratory Animals of the National Institutes of Health. $Chchd10^{S55L}$ knock-in mice were previously generated by CRISPR/Cas9 approach (Anderson et al, 2019) and are available as Stock #028952 from the Jackson Laboratory. Breeding was set up between $Chchd10^{S55L}$ heterozygous (Het) males with wildtype (WT) C57BL/6NJ females (Jackson Laboratory, stock #005304). Tail DNA was extracted with a Promega kit and genotype was determined by sequencing services at Transnetyx. Conditional heart frataxin (Fxn) exon-2 KO mice ($Fxn^{flox/null}$::MCK-Cre) were purchased from Jackson Laboratory (stock #029720).

## High-fat diet treatment

High-fat diet (HFD) containing 60% fat, 20% protein, 20% carbohydrates (D12492) and the calorie and sucrose matched control diet (CD; D12450J) were purchased from Research Diets (New Brunswick, NJ). For the pregnancy paradigm, Het females (P65) were mated with WT males and fed the HFD or CD for the mating/survival period until death. Offspring from the mating pairs were continued on the respective diet until adulthood. For FXN KO mice, treatment was started at 4 weeks of age.

## Echocardiography

Cardiac dimensions and function were analyzed by transthoracic echocardiogram using a Vevo 770 and 3100 Imaging Systems (VisualSonics, Toronto, ON, Canada) as previously described (Zhang et al, 2016). Briefly, male mice were lightly anesthetized with inhaled isoflurane (2% in $O_2$). Left-ventricle M-mode was used, and all measurements were obtained from the averaged values of three consecutive cardiac cycles. Left ventricle internal diameter end diastole (LVDd) and end systole (LVDs) were measured from the M-mode traces, and fractional shortening (FS) was calculated as follows: [(LVDd—LVDs)/LVDd]. Diastolic measurements were taken at the point of maximum cavity dimension, and systolic measurements were taken at the point of minimum cavity dimension, using the leading-edge method of the American Society of Echocardiography (Mitchell et al, 2019).

## Serum collection

Blood was collected by cheek bleeds into serum separator tubes. Serum was separated after 30 min of clotting at room temperature and centrifugation at $2000 \times g$ for 15 min. Serum liver enzymes were assessed at the Comparative Pathology Core at Memorial Sloan Kettering Cancer Center.

## Body weight and survival

Body weight was measured biweekly starting at weaning. For survival analyses, the age of death or the age at which institutional guidelines required euthanasia was recorded for the Kaplan-Meier survival curve.

## Metabolic phenotyping

Metabolic rate of Het and WT male mice was assessed in real time in the temperature and light-controlled Promethion High-definition Multiplexed Respirometry System at the WCM Metabolic Phenotyping Center. Food and water intake, and body mass are assessed gravimetrically. Distance traveled (activity) is determined by the XY position displacements represented by beam breaks. 24-h recordings were measured with and without a running wheel in the cage. Energy expenditure is determined from indirect calorimetry and calculated by the Weir equation, and RER is calculated as $VCO_2/VO_2$. Body composition, including total body, fat, and lean mass, as well as free and total water, in each mouse was measured by Echo-MRI.

## Intraperitoneal insulin tolerance test (IP-ITT)

Mice were fasted for 6 h prior to insulin injection. A bolus of 0.5 U/kg insulin was administered by IP. Tail vein blood was drawn at 15 min intervals from 0 to 90 min to assess glucose levels.

## Liver pathology

Pathological analyses were performed by the Laboratory of Comparative Pathology at Memorial Sloan Kettering Cancer Center. Female mice (aged P250-300) were sacrificed by cervical dislocation, tissues were harvested and fixed in 10% buffered formalin for 48 h. Liver was processed and embedded in paraffin. Sections were cut at a thickness of 5 µm and stained with hematoxylin and eosin (H&E). Quantification of fat accumulation

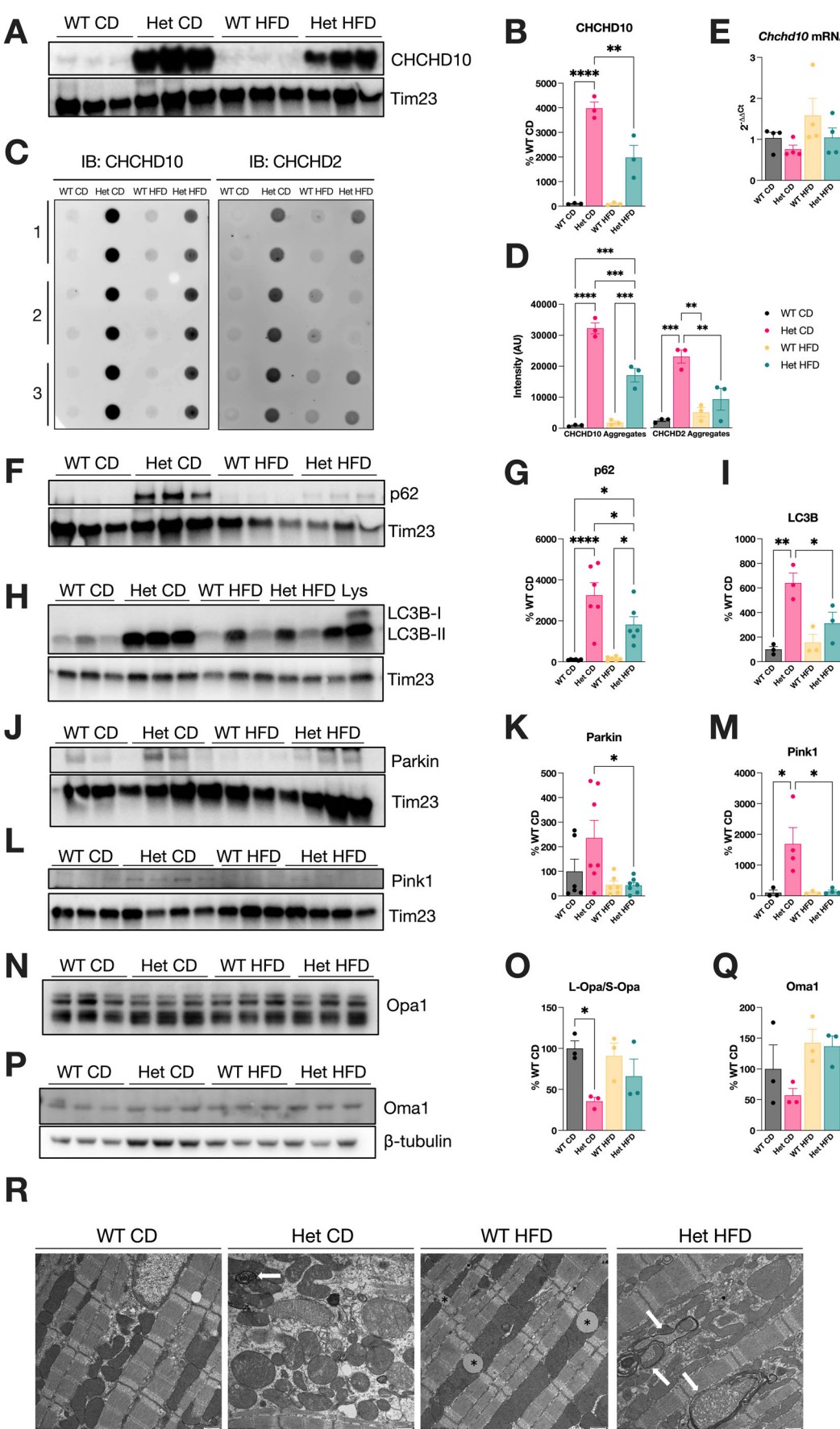

**Figure 8. HFD decreases protein aggregate burden and modulates mitophagy markers in Het hearts.**

(A) Representative western blot of CHCHD10 and mitochondrial inner membrane translocase (Tim23) in the heart of WT and Het mice. (B) Quantification of CHCHD10 in heart of WT and Het mice, normalized to Tim23 for mitochondrial content. (C) NP40-insoluble CHCHD10 and CHCHD2 aggregates trapped in nitrocellulose filter. IB, Immunoblot. (D) Quantification of NP40-insoluble aggregates. (E) Heart *Chchd10* mRNA normalized by β-actin mRNA and expressed as a percent of WT. (F) Representative western blot of P62 and Tim23 in the heart. (G) Quantification of P62, normalized to Tim23. (H) Representative western blot of LC3B and Tim23 in heart mitochondria, with 40 μg heart lysate (Lys) as a positive control. (I) Quantification of LC3B-II, normalized to Tim23. (J) Representative western blot of Parkin and Tim23. (K) Quantification of Parkin, normalized to Tim23. (L) Representative western blot of Pink1 and Tim23. (M) Quantification of Pink1, normalized to Tim23. (N) Representative western blot of Opa1 in heart homogenates. (O) Quantification of Long Opa1 to the proteolytically processed short Opa1. (P) Representative western blot of Oma1 on heart homogenates. (Q) Quantification of Oma1, normalized to β-tubulin. (R) Electron microscopy of hearts at P300. Asterisks denote lipid droplets associated with mitochondria. Arrows indicate abnormal mitochondria engulfed by phagosome membranes. Scale bar = 500 nm. Data information: For (A–D), (H), (I), (L–Q), and (P), WT CD ($n = 3$ mice), Het CD ($n = 3$ mice), WT HFD ($n = 3$ mice), Het HFD ($n = 3$ mice). For (E), WT CD ($n = 4$ mice), Het CD ($n = 4$ mice), WT HFD ($n = 4$ mice), Het HFD ($n = 4$ mice) with two technical replicates per sample. For (F), (G), (J), and (K), WT CD ($n = 6$ mice), Het CD ($n = 6$ mice), WT HFD ($n = 6$ mice), Het HFD ($n = 6$ mice). For (B), (D), (E), (G), (I), (K), (M), (O), and (Q), statistical significance was determined by one-way ANOVA with Tukey's correction for multiple comparisons with samples normalized to WT CD. *$p < 0.05$, **$p < 0.01$, ***$p < 0.0005$, ****$p < 0.0001$. Data are expressed as mean ± SEM. Source data are available online for this figure.

in the liver was performed with ImageJ by thresholding images of sections taken at 20× on a light microscope so that the empty optical areas corresponding to lipid droplets could be defined and measured. The area of droplets relative to the total area of each field was measured. Quantification was performed on 14 image fields from 2 mice per group. Quantification was done by an experimenter who was blinded to both genotype and treatment.

## Heart electron microscopy

Mice were terminally anesthetized with sodium pentobarbital (150 mg/kg, i.p.) and perfused intracardially with 2% heparin in normal saline followed by 3.75% acrolein and 2% PFA in phosphate buffer (PB). Tissues were post-fixed in 2% acrolein and 2% PFA in PB for 30 min. Heart sections were post-fixed in 2% osmium tetroxide, dehydrated, and flat embedded. Ultrathin sections (70 nm) were cut, collected on copper grids, counterstained with uranyl acetate and Reynold's lead citrate, and imaged on an electron microscope (CM10, FEI).

## Western Blotting

Mitochondrial-enriched fractions were isolated from heart tissue by differential centrifugation (Gostimskaya and Galkin, 2010). Protein concentration was determined by the Bradford protein assay (Bio-Rad). Mitochondrial (5 μg) fractions and heart homogenates (25 μg) were denatured in 1X Laemmli Buffer (Bio-Rad) containing 2-Mercaptoethanol at 95 °C for 10 min and separated by electrophoresis in Any kD™ Mini-PROTEAN TGX precast gel (Bio-Rad) and transferred to a PVDF membrane (Bio-Rad). For the Opa1, Myh6, and PMCA blots, a 4–12% NuPAGE™ Bis-Tris protein gel (Thermo Fisher) with the MOPS running buffer was used for electrophoresis. Blots were incubated in 3% BSA in TBS with 1% Tween-20 (TBST) for one hour at room temperature. Primary antibodies were incubated overnight at 4 °C. Secondary antibodies were incubated for 45 min at room temperature. In all blots, proteins were detected using Clarity Western ECL Blotting Substrates (Bio-Rad) and imaged on ChemiDoc Touch (Bio-Rad). The antibodies used are anti-CHCHD10 (1:1000, Proteintech, 25671-AP), anti-p62/SQSTM1 (1:1000, Novus, H00008878-M01), anti-Parkin (1:1000, Santa Cruz, SC32282), anti-LC3B (1:1000, Cell Signaling, 2775) anti-OPA1 (1:1000, BD Transduction Lab, 612606), anti-Oma1 (1:1000, Proteintech, 17116-1-AP), anti-Pink1 (1:1000, Novus, BC100-494), anti-Myh6 (1:25,000, Proteintech, 22281-1-AP), anti-PMCA (1:2000, Thermofisher, MA3-914),

anti-α-actin (1:10,000, Proteintech, 23660-1-AP) and anti-β-tubulin (1:2000, Thermofisher, #MA5-16308).

## Filter trap assay

Insoluble protein aggregates were detected by filter trap assay as previously described (Palomo et al, 2018). Briefly, 5 μg of mitochondrial fractions were solubilized with 1% NP-40 in PBS for 15 min on ice. Samples were loaded onto a Bio-Dot Microfiltration apparatus (Bio-Rad) containing a cellulose acetate membrane (0.2 μm pore diameter, Whatman). Vacuum was applied to pass samples through the membrane, which was then washed with 1% Tween-20 in PBS. Proteins were detected with either anti-CHCHD10 antibody (1:500, Proteintech, 25671-AP) or anti-CHCHD2 antibody (1:500, Proteintech, 19424-1-AP) followed by donkey anti-rabbit secondary antibody (Licor, 926-68073). Blots were then imaged on the Odyssey CLx (Licor).

## 3′-RNA seq

RNA was extracted from heart (P75) using TRIzol (Thermo Fisher) and the SV Total RNA Isolation System (Promega). 3′RNAseq libraries were prepared from 500 ng of RNA per sample using the Lexogen QuantSeq 3′ mRNA-Seq Library Prep Kit FWD for Illumina and pooled for reduced run variability. Libraries were sequenced with single-end 86 bps on an Illumina NextSeq500 sequencer (Cornell BRC Facility). Raw sequence reads were processed using the BBDuk program in the BBMap package. Trimmed reads were aligned to the mouse genome assembly GRCm38.p6 using the STAR aligner (version 2.5.0a). SAM files were converted to BAM to read overlapping reads per gene using HTSeq-count (version 0.6.1). Gene expression profiles were constructed for differential expression, cluster, and principal component analyses with the R package DESeq2 (Auranen et al, 2015). For normalization and differential gene expression analysis, a low counts filter of <96 was used, and all other filtering parameters were kept as defaults. A Wald test was used to determine statistical significance, with the cutoff being a False Discovery Rate <5% after Benjamini–Hochberg correction. Pathway analysis for all gene expression data was performed with the gprofiler2 (Kolberg et al, 2020) and clusterProfiler packages (Wu et al, 2021), using the gene ontology (GO) Molecular Function (GO:MF), GO Biological Process (GO:BP), and Kyoto Encyclopedia of Genes and Genomes (KEGG) databases. Cutoff for significance

was FDR corrected $p$-value < 0.05. Pathways shown in the figures were condensed using the simplify function from the clusterProfiler package (Wu et al, 2021) to merge terms with more than 40% overlapping annotated genes.

## Quantitative real-time PCR

RNA was extracted from heart using Trizol-chloroform precipitation and the RNeasy Mini Kit (Qiagen). cDNA was generated using the ImProm-II Reverse Transcription System (Promega) according to manufacturer's instructions. Quantitative real-time PCR (qPCR) amplification of cDNA was performed with the SYBR Green PCR Master Mix (ThermoFisher Scientific) on a QuantStudio 6 Flex Real-Time PCR system (ThermoFisher Scientific). Standard cycling parameters were used, and relative expression of the genes of interest was normalized to $\beta$-actin. mRNA expression levels were quantified using the following primers: *Chchd10* forward: 5'-CACTCAGAGCGACCTAACCC-3', reverse: 5'-GGAGCTCA-GACCGTGATTGT-3'; *β-actin* forward: 5'-CTTTGCAGCTCC TCCGTTGC-3', reverse: 5'-CCTTCTGACCCATTCCCACC-3'.

## Droplet PCR for mtDNA copy number

Genomic heart DNA was isolated by phenol-chloroform extraction (Bacman et al, 2010). Droplet PCR (dPCR) was performed on the QIAcuity dPCR System (QIAcuityOne, 5plex device-ID: 911021) for mtDNA copy number analyses. Cycling parameters were as previously described (Bacman et al, 2024). The mtDNA copy number was quantified by the ratio of mtND5 (mtDNA reference) and 18s rRNA (nuclear reference) using the following primers and probes: mtND5 forward: 5′-CCTGAGCCCTACTAATTACAC, mtDNA reverse: 5′-GAGATGACAAATCCTGCAAAG, mtND5 probe: 5'/HEX/ACCCAATCAAACGCCTAGCATTCG/-3'; 18s rRNA forward: 5′-CGTCTGCCCTATCAACTTT-3', 18s rRNA reverse: 5′-CCTCGAAAGAGTCCTGTATTG-3', 18s rRNA probe: 5′-/5Cy5/AGAAACGGCTACCACATCC/3IAbRQSp/-3′.

## Untargeted and targeted polar metabolomic profiling

15 mg of cardiac tissue was homogenized in 80% methanol (Sigma) using Tissue Tearer (BioSpec) on dry ice. Samples were incubated at −80 °C for 4 h. For serum, 80% methanol was added. Homogenates and serum were then centrifuged at 14,000 rfc for 20 min at 4 °C. The supernatant was extracted and stored at −80 °C. The Weill Cornell Medicine Meyer Cancer Center Proteomics & Metabolomics Core Facility performed hydrophilic interaction liquid chromatography-mass spectrometry (LC-MS) for relative quantification of polar metabolite profiles. Metabolites were measured on a Q Exactive Orbitrap mass spectrometer, coupled to a Vanquish UPLC system by an Ion Max ion source with a HESI II probe (Thermo Scientific). A Sequant ZIC-pHILIC column (2.1 mm i.d. × 150 mm, particle size of 5 µm, Millipore Sigma) was used for separation. The MS data was processed using XCalibur 4.1 (Thermo Scientific) to obtain the metabolite signal intensity for relative quantitation. Targeted identification was available for 205 metabolites based on an in-house library established using known chemical standards. Identification required exact mass (within 5 ppm) and standard retention times. For untargeted metabolomics, metabolites were identified by mass

matching of the MS signal to metabolites in the HMDB database. If multiple metabolites in the database were matched to a certain MS signal, all matched metabolites were grouped into a single identification, and ordered based on the number of references included in the HMDB database (high to low). We used the first ranked metabolite in the downstream analyses. When multiple values with the same metabolite attribution occurred (different metabolites with same mass and retention time), we opted to use all values in the analyses to avoid biases on which intensities/attributions to consider. Peak intensities for metabolites were screened for missing values and relative metabolite abundance data was analyzed by using MetaboAnalyst software version 5.0 (Pang et al, 2021). Metabolite significance was determined with one-way ANOVA with post-hoc t-tests, with the cutoff being a raw $p$ value < 0.05, and the pathway significance cutoff was FDR corrected $p$ value < 0.05.

## Lipidomics

Lipids were extracted from samples as previously described (Satomi et al, 2017). In brief, 1 ml or 180 µl of 90% isopropanol was added to 15 mg of homogenized heart tissue or 20 µl serum, respectively. Samples were vortexed for 2 min, sonicated for 1 min using a probe sonicator and centrifuged at 15,000 g for 10 min. The supernatant was collected and dried down using a SpeedVac. The dried samples were reconstituted using acetonitrile/isopropanol/water 65:30:5 containing stable isotope-labeled internal lipid standards (Splash Lipidomix, Avanti Polar Lipids) prior to LC-MS analysis. Chromatographic separation was performed on a Vanquish UHPLC system with a Cadenza CD-C18 3 µm packing column (Imtakt, 2.1 mm id × 150 mm) coupled to a Q Exactive Orbitrap mass spectrometer (Thermo Scientific) via an Ion Max ion source with a HESI II probe (Thermo Scientific). The mobile phase consisted of buffer A: 60% acetonitrile, 40% water, 10 mM ammonium formate with 0.1% formic acid and buffer B: 90% isopropanol, 10% acetonitrile, 10 mM ammonium formate with 0.1% formic acid. The LC gradient was as follows: 0–1.5 min, 32% buffer B; 1.5–4 min, 32–45% buffer B; 4–5 min, 45–52% buffer B; 5–8 min, 52–58% buffer B; 8–11 min, 58–66% buffer B; 11–14 min, 66–70% buffer B; 14–18 min, 70–75% buffer B; 21–25 min, isocratic 97% buffer B, 25–25.1 min 97–32% buffer B; followed by 5 min of re-equilibration of the column before the next run. The flow rate was 200 µl/min. A data-dependent mass spectrometric acquisition method was used for lipid identification. In this method, each MS survey scan was followed by up to 10 MS/MS scans performed on the most abundant ions. Data was acquired in positive and negative mode in separate runs. The following electrospray parameters were used: spray voltage 3.0 kV, heated capillary temperature 350 °C, HESI probe temperature 350 °C, sheath gas, 35 units; auxiliary gas 10 units. For MS scans: resolution, 70,000 (at $m/z$ 200); automatic gain control target, 3e6; maximum injection time, 200 ms; scan range, 250–1800 $m/z$. For MS/MS scans: resolution, 17,500 (at 200 $m/z$); automatic gain control target, 1e5 ions; maximum injection time, 75 ms; isolation window, 1 $m/z$; NCE, stepped 20,30 and 40.

The LC-MS results were processed using MS-DIAL software (version 4.9) for lipid identification and relative quantitation.

## Free fatty acid analyses

Free fatty acids were extracted from serum using 90% methanol (LC/MS grade, Thermo Scientific). The extracts were clarified by

**The paper explained**

**Problem**

Coiled-helix-coiled-helix domain containing 10 (CHCHD10) mutations cause autosomal dominant diseases, including fatal mitochondrial cardiomyopathy. Research has shown that CHCHD10 diseases involve proteotoxic mitochondrial stress and metabolic rewiring but approaches to modify these pathogenic mechanisms have not been tested.

**Results**

Administration of a high-fat diet in a mouse model of CHCHD10 mitochondrial cardiomyopathy increases fatty acid utilization and decreases CHCHD10 aggregate burden by enhancing mitophagy. High-fat diet downregulates key cardiomyopathy markers, improves heart function, and increases survival of mutant mice.

**Impact**

Our findings indicate that metabolic therapeutic strategies could be beneficial in mitochondrial cardiomyopathies. High-fat diet enhances the utilization of fatty acids, which is the preferred metabolic fuel of the adult healthy heart. This metabolic rearrangement attenuates the maladaptive increase in glucose utilization and one-carbon metabolism. Furthermore, high-fat diet promotes turnover of mitochondria containing CHCHD10 aggregates, thereby decreasing mitochondrial stress.

All numerical data are expressed as mean ± standard error of the mean. Statistical comparisons were made in GraphPad Prism. For metabolite comparisons, ANOVA with Tukey's correction for multiple comparisons was used. Statistical significance was considered at $p < 0.05$ (*$p < 0.05$, **$p < 0.01$, ***$p < 0.0005$, ****$p < 0.0001$). For survival studies, data were analyzed by Kaplan–Meir curve and Mantel-Cox log-rank test. Biological replicate information for each experiment, as well as number of mice, age, and sex, are indicated in the figure legends.

## Data availability

The source data of transcriptomics and metabolomics in this study has been deposited. Datasets are available in the following databases: GEO: Gene Expression Omnibus GSE261796. NMDR: National Metabolomics Data Repository (Sud et al, 2016) PR001944 (https://doi.org/10.21228/M8FD9G).

The source data of this paper are collected in the following database record: biostudies:S-SCDT-10_1038-S44321-024-00067-5.

## Peer review information

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

centrifugation at $15,000 \times g$ for 10 min. The supernatant was collected and dried down using a SpeedVac. The dried sample was reconstituted using 50% methanol prior to LC-MS analysis. Chromatographic separation was performed on a Vanquish UHPLC system (Thermo Scientific) with a Cadenza CD-C18 3 μm packing column (Imtakt, 2.1 mm id × 150 mm) coupled to a Q Exactive Orbitrap mass spectrometer (Thermo Scientific) via an Ion Max ion source with a HESI II probe (Thermo Scientific). The mobile phase consisted of buffer A: 5 mM ammonium acetate in water (LC/MS grade, Thermo Scientific) and buffer B: 5 mM ammonium acetate, 85% isopropanol (LC/MS grade, Thermo Scientific), 10% acetonitrile (LC/MS grade, Thermo Scientific), and 5% water. The LC gradient was as follows: 0–1.5 min, 50% buffer B; 1.5–3 min, 50–90% buffer B; 3–5.5 min, 90–95% buffer B; 5.5–10 min, 95% buffer B, followed by 5 min of re-equilibration of the column before the next run. The flow rate was 150 μL/min. MS data was acquired in negative mode. The following electrospray parameters were used: spray voltage 3.0 kV, heated capillary temperature 350 °C, HESI probe temperature 350 °C, sheath gas, 35 units; auxiliary gas 10 units. For MS scans: mass scan range, 140–1000 $m/z$; resolution, 70,000 (at $m/z$ 200); automatic gain control target, 1e6; maximum injection time, 50 ms. MS data files were processed using XCalibur (version 4.1, Thermo Scientific). Identification of free fatty acids was based on accurate masses within 5 ppm and standard retention times from compound standards. Relative quantitation was performed based on MS signal intensities.

### Experimental design and statistical analyses

Unless specified, experimenters were not blinded due to the nature of the experiments. Data visualization was done in R using the ggplot2, Enhanced volcano plot, and Venn diagram packages available from CRAN, and in GraphPad Prism version 9.3 (GraphPad Software, Inc).

Fessler E, Eckl EM, Schmitt S, Mancilla IA, Meyer-Bender MF, Hanf M, Philippou-Massier J, Krebs S, Zischka H, Jae LT (2020) A pathway coordinated by DELE1 relays mitochondrial stress to the cytosol. Nature 579:433–437

Funai K, Summers SA, Rutter J (2020) Reign in the membrane: how common lipids govern mitochondrial function. Curr Opin Cell Biol 63:162–173

Genin EC, Bannwarth S, Lespinasse F, Ortega-Vila B, Fragaki K, Itoh K, Villa E, Lacas-Gervais S, Jokela M, Auranen M et al (2018) Loss of MICOS complex integrity and mitochondrial damage, but not TDP-43 mitochondrial localisation, are likely associated with severity of CHCHD10-related diseases. Neurobiol Dis 119:159–171

Genin EC, Bannwarth S, Ropert B, Lespinasse F, Mauri-Crouzet A, Auge G, Fragaki K, Cochaud C, Donnarumma E, Lacas-Gervais S et al (2022) CHCHD10 and SLP2 control the stability of the PHB complex: a key factor for motor neuron viability. Brain 145:3415–3430

Genin EC, Madji Hounoum B, Bannwarth S, Fragaki K, Lacas-Gervais S, Mauri-Crouzet A, Lespinasse F, Neveu J, Ropert B, Auge G et al (2019) Mitochondrial defect in muscle precedes neuromuscular junction degeneration and motor neuron death in CHCHD10(S59L/+) mouse. Acta Neuropathol 138:123–145

Gostimskaya I, Galkin A (2010) Preparation of highly coupled rat heart mitochondria. J Vis Exp 43:2202

Guo X, Aviles G, Liu Y, Tian R, Unger BA, Lin YT, Wiita AP, Xu K, Correia MA, Kampmann M (2020) Mitochondrial stress is relayed to the cytosol by an OMA1-DELE1-HRI pathway. Nature 579:427–432

Johnson JO, Glynn SM, Gibbs JR, Nalls MA, Sabatelli M, Restagno G, Drory VE, Chio A, Rogaeva E, Traynor BJ (2014) Mutations in the CHCHD10 gene are a common cause of familial amyotrophic lateral sclerosis. Brain 137:e311

Kim JA, Wei Y, Sowers JR (2008) Role of mitochondrial dysfunction in insulin resistance. Circ Res 102:401–414

Kitakaze M, Hori M (2000) Adenosine therapy: a new approach to chronic heart failure. Expert Opin Investig Drugs 9:2519–2535

Kolberg L, Raudvere U, Kuzmin I, Vilo J, Peterson H (2020) gprofiler2—an R package for gene list functional enrichment analysis and namespace conversion toolset g:Profiler. F1000Res 9:ELIXER-709

Kuhl I, Miranda M, Atanassov I, Kuznetsova I, Hinze Y, Mourier A, Filipovska A, Larsson NG (2017) Transcriptomic and proteomic landscape of mitochondrial dysfunction reveals secondary coenzyme Q deficiency in mammals. Elife 6:e30952

Liu T, Wetzel L, Zhu Z, Kumaraguru P, Gorthi V, Yan Y, Bukhari MZ, Ermekbaeva A, Jeon H, Kee TR et al (2023) Disruption of mitophagy flux through the PARL-PINK1 pathway by CHCHD10 mutations or CHCHD10 depletion. Cells 12:2781

Liu YT, Huang X, Nguyen D, Shammas MK, Wu BP, Dombi E, Springer DA, Poulton J, Sekine S, Narendra DP (2020) Loss of CHCHD2 and CHCHD10 activates OMA1 peptidase to disrupt mitochondrial cristae phenocopying patient mutations. Hum Mol Genet 29:1547–1567

Mitchell C, Rahko PS, Blauwet LA, Canaday B, Finstuen JA, Foster MC, Horton K, Ogunyankin KO, Palma RA, Velazquez EJ (2019) Guidelines for performing a comprehensive transthoracic echocardiographic examination in adults: recommendations from the American Society of Echocardiography. J Am Soc Echocardiogr 32:1–64

Muller K, Andersen PM, Hubers A, Marroquin N, Volk AE, Danzer KM, Meitinger T, Ludolph AC, Strom TM, Weishaupt JH (2014) Two novel mutations in conserved codons indicate that CHCHD10 is a gene associated with motor neuron disease. Brain 137:e309

Nguyen MK, McAvoy K, Liao SC, Doric Z, Lo I, Li H, Manfredi G, Nakamura K (2021) Mouse midbrain dopaminergic neurons survive loss of the PD-associated mitochondrial protein CHCHD2. Hum Mol Genet 31:1500–1518

Palomo GM, Granatiero V, Kawamata H, Konrad C, Kim M, Arreguin AJ, Zhao D, Milner TA, Manfredi G (2018) Parkin is a disease modifier in the mutant SOD1 mouse model of ALS. EMBO Mol Med 10:e8888

Pang Z, Chong J, Zhou G, de Lima Morais DA, Chang L, Barrette M, Gauthier C, Jacques P, Li S, Xia J (2021) MetaboAnalyst 5.0: narrowing the gap between raw spectra and functional insights. Nucleic Acids Res 49:W388–w396

Penttila S, Jokela M, Bouquin H, Saukkonen AM, Toivanen J, Udd B (2015) Late onset spinal motor neuronopathy is caused by mutation in CHCHD10. Ann Neurol 77:163–172

Prasun P, LoPiccolo MK, Ginevic I (1993) Long-chain hydroxyacyl-CoA dehydrogenase deficiency/trifunctional protein deficiency. In: GeneReviews(®), Adam MP, Everman DB, Mirzaa GM, Pagon RA, Wallace SE, Bean LJH, Gripp KW, Amemiya A (eds.) University of Washington, Seattle

Puccio H, Simon D, Cossee M, Criqui-Filipe P, Tiziano F, Melki J, Hindelang C, Matyas R, Rustin P, Koenig M (2001) Mouse models for Friedreich ataxia exhibit cardiomyopathy, sensory nerve defect and Fe-S enzyme deficiency followed by intramitochondrial iron deposits. Nat Genet 27:181–186

Ritterhoff J, Tian R (2017) Metabolism in cardiomyopathy: every substrate matters. Cardiovasc Res 113:411–421

Rotig A, de Lonlay P, Chretien D, Foury F, Koenig M, Sidi D, Munnich A, Rustin P (1997) Aconitase and mitochondrial iron-sulphur protein deficiency in Friedreich ataxia. Nat Genet 17:215–217

Satomi Y, Hirayama M, Kobayashi H (2017) One-step lipid extraction for plasma lipidomics analysis by liquid chromatography mass spectrometry. J Chromatogr B Anal Technol Biomed Life Sci 1063:93–100

Sayles NM, Southwell N, McAvoy K, Kim K, Pesini A, Anderson CJ, Quinzii C, Cloonan S, Kawamata H, Manfredi G (2022) Mutant CHCHD10 causes an extensive metabolic rewiring that precedes OXPHOS dysfunction in a murine model of mitochondrial cardiomyopathy. Cell Rep 38:110475

Shammas MK, Huang X, Wu BP, Fessler E, Song IY, Randolph NP, Li Y, Bleck CK, Springer DA, Fratter C et al (2022) OMA1 mediates local and global stress responses against protein misfolding in CHCHD10 mitochondrial myopathy. J Clin Invest 132:e157504

Sud M, Fahy E, Cotter D, Azam K, Vadivelu I, Burant C, Edison A, Fiehn O, Higashi R, Nair KS et al (2016) Metabolomics Workbench: An international repository for metabolomics data and metadata, metabolite standards, protocols, tutorials and training, and analysis tools. Nucleic Acids Res 44:D463–470

Tan Y, Li M, Wu G, Lou J, Feng M, Xu J, Zhou J, Zhang P, Yang H, Dong L et al (2021) Short-term but not long-term high fat diet feeding protects against pressure overload-induced heart failure through activation of mitophagy. Life Sci 272:119242

Tong M, Saito T, Zhai P, Oka SI, Mizushima W, Nakamura M, Ikeda S, Shirakabe A, Sadoshima J (2019) Mitophagy is essential for maintaining cardiac function during high fat diet-induced diabetic cardiomyopathy. Circ Res 124:1360–1371

Vangaveti V, Baune BT, Kennedy RL (2010) Hydroxyoctadecadienoic acids: novel regulators of macrophage differentiation and atherogenesis. Ther Adv Endocrinol Metab 1:51–60

Wai T, Garcia-Prieto J, Baker MJ, Merkwirth C, Benit P, Rustin P, Ruperez FJ, Barbas C, Ibanez B, Langer T (2015) Imbalanced OPA1 processing and mitochondrial fragmentation cause heart failure in mice. Science 350:aad0116

Wu T, Hu E, Xu S, Chen M, Guo P, Dai Z, Feng T, Zhou L, Tang W, Zhan L et al (2021) clusterProfiler 4.0: a universal enrichment tool for interpreting omics data. Innovation 2:100141

Xia W, Qiu J, Peng Y, Snyder MM, Gu L, Huang K, Luo N, Yue F, Kuang S (2022) Chchd10 is dispensable for myogenesis but critical for adipose browning. Cell Regen 11:14

Zhang Y, Huang Y, Cantalupo A, Azevedo PS, Siragusa M, Bielawski J, Giordano FJ, Di Lorenzo A (2016) Endothelial Nogo-B regulates sphingolipid biosynthesis to promote pathological cardiac hypertrophy during chronic pressure overload. JCI Insight 1:e85484

## Acknowledgements

We thank the WCM Proteomics & Metabolomics Core Facility, Metabolic Phenotyping Core, Comparative Pathology Core at Memorial Sloan Kettering Cancer Center, the WCM Neuroanatomy and Electron Microscopy Core, CLC Microscope and Image Analysis Core, and the Cornell University BRC Genomics Facility for their contributions. Schematics were created with BioRender.com. NextGenALS and Project ALS 2021-01 to HK and GM, Muscular Dystrophy Association (MDA) 961871-01 to HK, NIH/NINDS R35 NS122209 to GM, MDA602894 to GM, NIH/NINDS NS122209-02S1 to NS, NIH grant U2C-DK119886 and OT2-OD030544 to the NIH Common Fund's National Metabolomics Data Repository (NMDR).

## Author contributions

**Nneka Southwell**: Data curation; Formal analysis; Investigation; Methodology; Writing—original draft; Writing—review and editing. **Onorina Manzo**: Data curation; Formal analysis; Investigation; Methodology; Writing—original draft. **Sandra Bacman**: Data curation; Formal analysis; Investigation; Methodology. **Dazhi Zhao**: Data curation; Investigation; Methodology. **Nicole M Sayles**: Data curation; Formal analysis; Investigation; Visualization; Methodology. **Jalia Dash**: Data curation; Methodology. **Keigo Fujita**: Formal analysis. **Marilena D'Aurelio**: Supervision. **Annarita Di Lorenzo**: Data curation; Formal analysis; Supervision; Investigation; Methodology. **Giovanni Manfredi**: Conceptualization; Data curation; Formal analysis; Supervision; Funding acquisition; Visualization; Writing—original draft; Project administration; Writing—review and editing. **Hibiki Kawamata**: Conceptualization; Data curation; Formal analysis; Supervision; Funding acquisition; Visualization; Writing—original draft; Project administration; Writing—review and editing.

Source data underlying figure panels in this paper may have individual authorship assigned. Where available, figure panel/source data authorship is listed in the following database record: biostudies:S-SCDT-10_1038-S44321-024-00067-5.

## Disclosure and competing interests statement

The authors declare no competing interests.

# Expanded View Figures

**A**

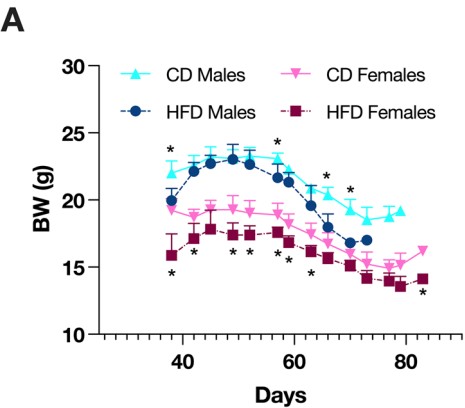

**B**

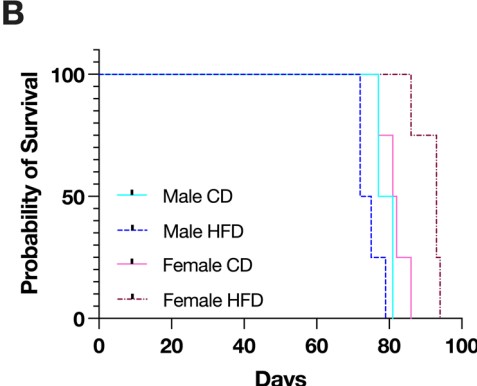

**C**

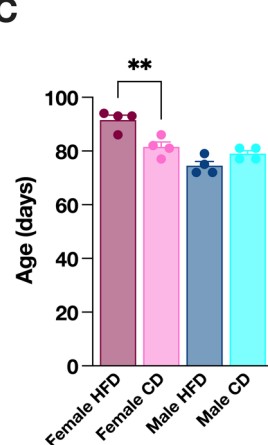

**Figure EV1.  Effect of HFD in *Fxn* KO mice hearts.**

(A) Body weight of cardiac *Fxn* KO mice on CD or HFD. (B) Kaplan–Meir survival curve of *Fxn* KO mice on CD or HFD. (C) Mean age at death of heart *Fxn* KO mice on CD or HFD. Data information: In (A–C), *Fxn* KO CD males ($n = 4$ mice), *Fxn* KO CD females ($n = 4$ mice), *Fxn* KO HFD males ($n = 4$ mice), *Fxn* KO HFD females ($n = 4$ mice). For (A) and (C), statistical significance was determined by unpaired t-tests within each sex. *$p < 0.05$, **$p < 0.01$. Data are expressed as mean ± SEM. Source data are available online for this figure.

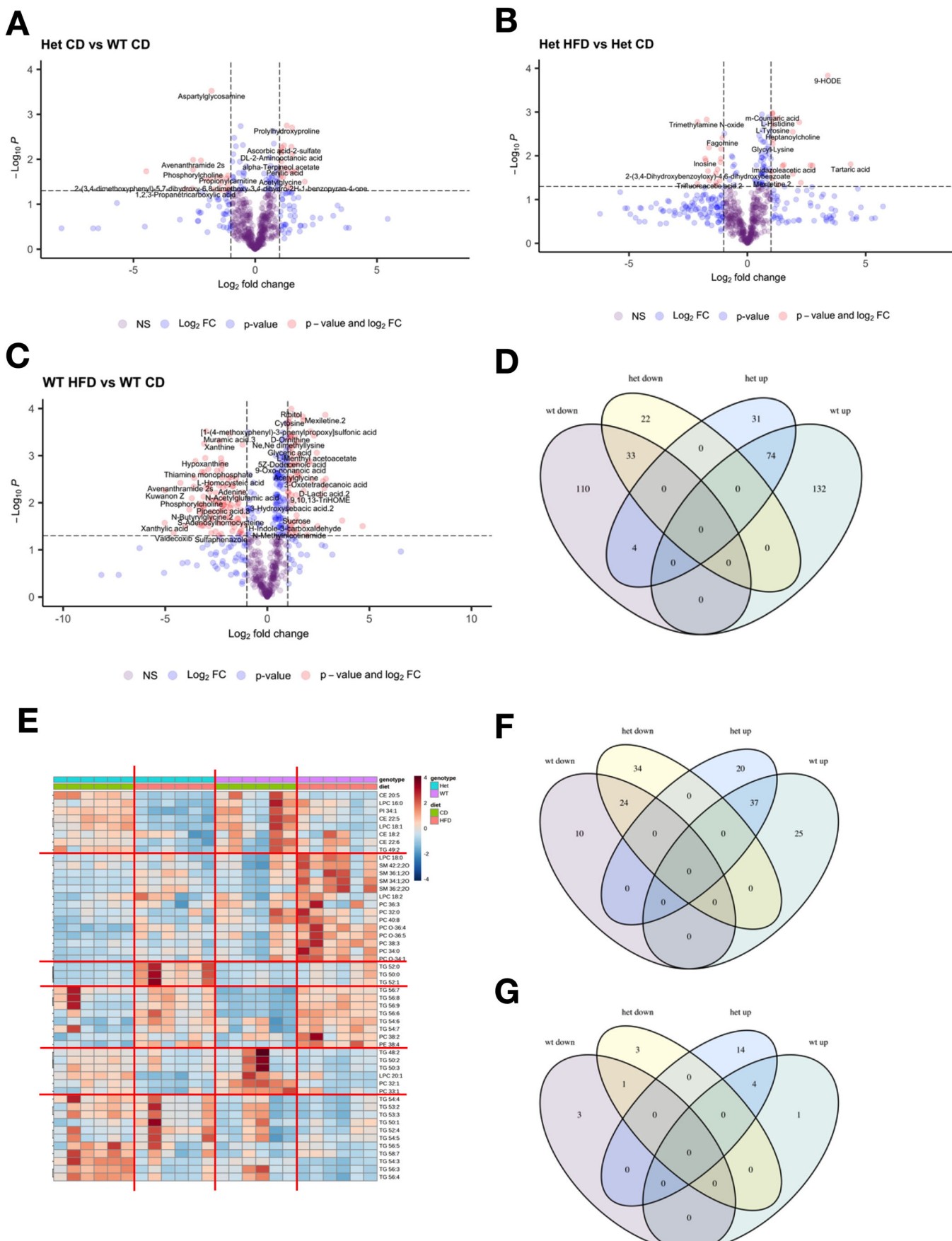

◀

**Figure EV2. Blood metabolomics analyses.**

(A) Volcano plot of serum metabolites in Het CD and WT CD mice at P250. (B) Volcano plot of serum metabolites in Het HFD and Het CD mice. (C) Volcano plot of serum metabolites in WT HFD and WT CD mice. (D) Venn diagram of serum metabolites both up- and down-regulated by HFD within each genotype separately (Het HFD vs Het CD, WT HFD vs WT CD). (E) Heatmap of top 50 metabolites (by interquartile range) in serum of Het and WT mice on CD or HFD. (F) Venn diagram of serum lipids up- or down-regulated by HFD within each genotype separately. (G) Venn diagram of serum free fatty acids up- or down-regulated by HFD within each genotype separately. Data information: In (A–G), female groups, WT CD ($n = 6$ mice), Het CD ($n = 6$ mice), WT HFD ($n = 6$ mice), Het HFD ($n = 6$ mice). For (A–C), the threshold for volcano plots were set at $\log_2$FC $= 1$ and $p_{raw} < 0.05$. For (D), (F) and (G), the Venn diagram threshold was $p < 0.1$. For (E), the red lines and the numbers on the left denote different groups of metabolites that change based on genotype and diet.

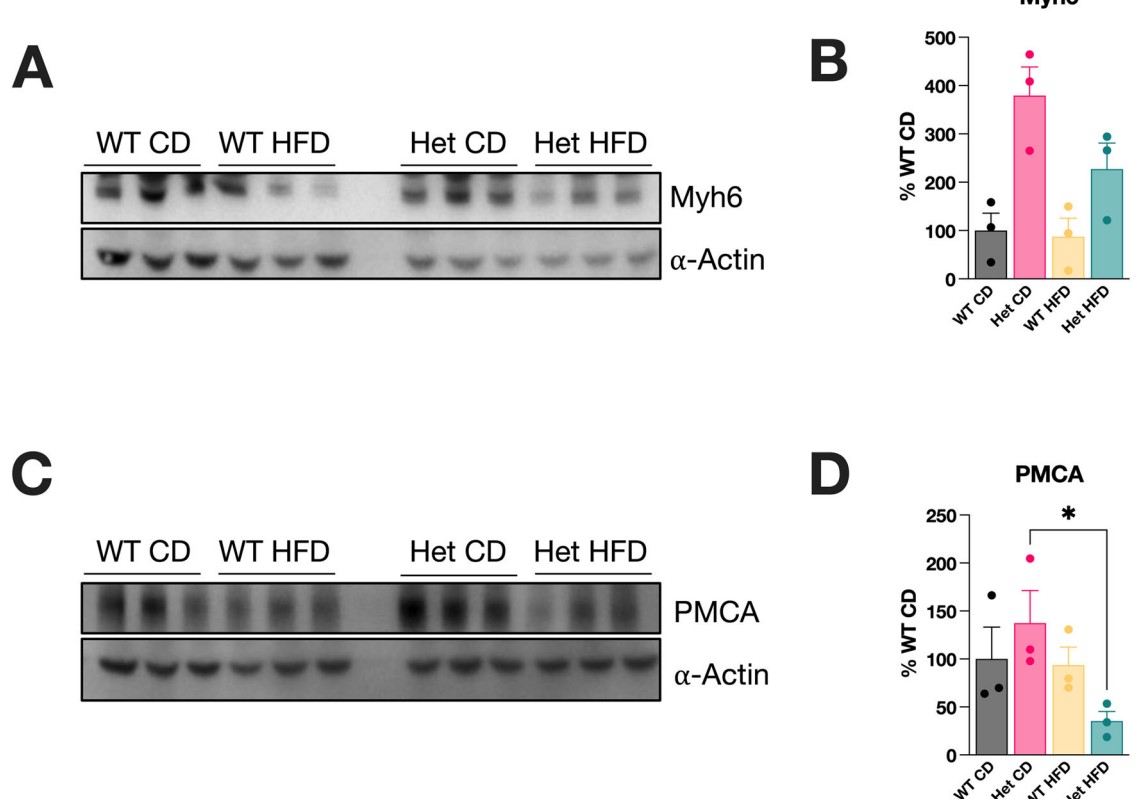

**Figure EV3.  Decreased protein levels of cardiomyopathy stress markers in Het HFD hearts.**

(A) Representative western blot image of Myh6 normalized to α-actin. (B) Quantification of Myh6 protein levels normalized to α-actin (C) Representative western blot image of PMCA (*Atp2b4*) normalized to α-actin. (D) Quantification of PMCA protein levels normalized to α-actin. Data information: In (A–D), WT CD (*n* = 3 mice), Het CD (*n* = 3 mice), WT HFD (*n* = 3 mice), Het HFD (*n* = 3 mice). For (B) and (D), data are expressed as mean ± SEM. Statistical significance was determined by unpaired t-test between genotypes. *$p$ < 0.05. Source data are available online for this figure.

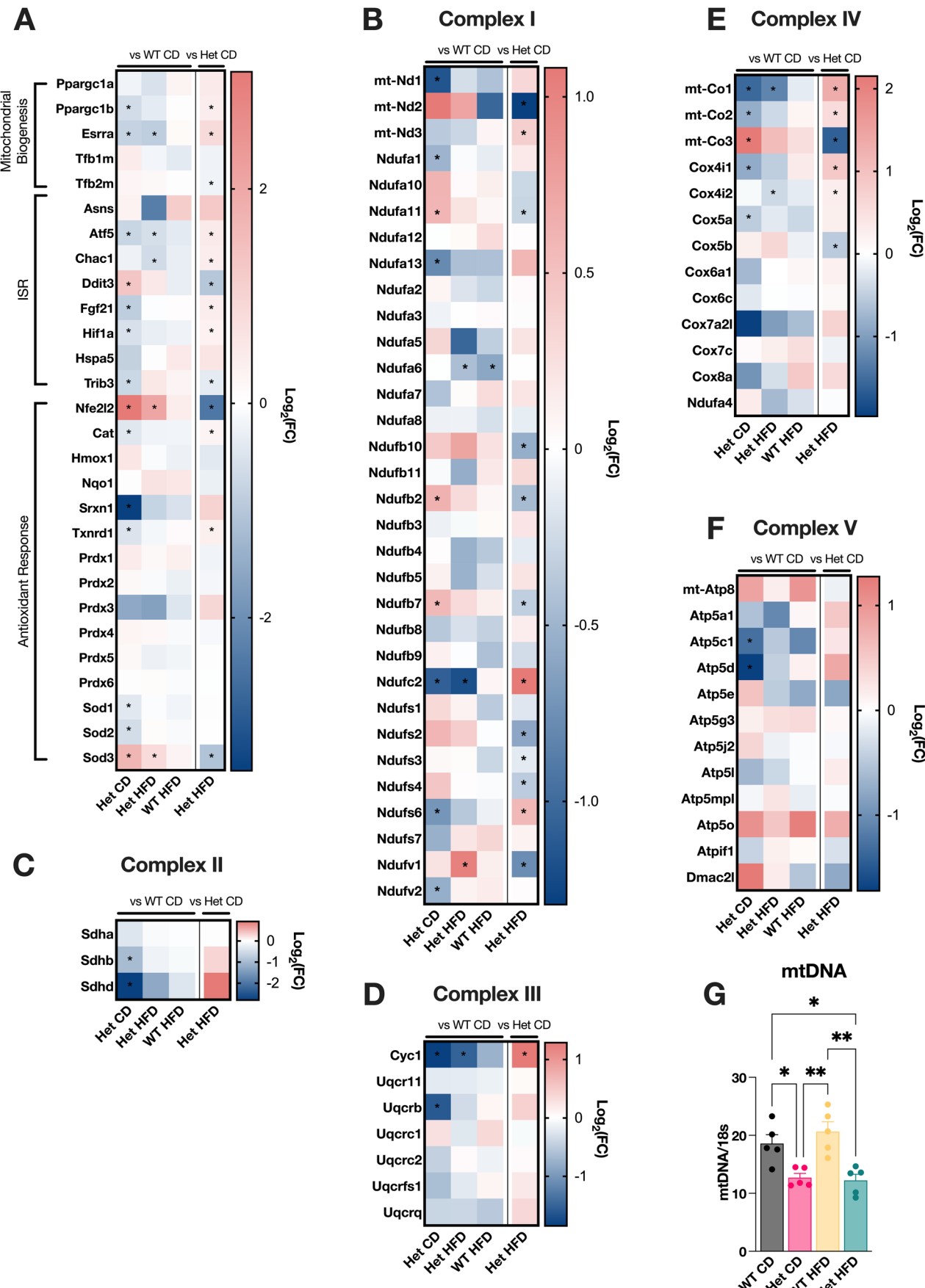

**Figure EV4.  Heart transcriptomics of mitochondrial respiratory complexes, mitochondrial biogenesis, ISR, and antioxidant genes.**

(A) Heatmap with genes related to mitochondrial biogenesis, ISR, and antioxidant responses. (B) Heatmap showing effect of HFD on genes related to Complex I. (C) Heatmap with Complex II genes. (D) Heatmap with Complex III genes. (E) Heatmap with Complex IV genes. (F) Heatmap with Complex V genes. (G) Heart mtDNA/nDNA (ND5/18s rRNA) ratio by dPCR. Data information: In (A–F), WT CD ($n = 10$ mice), Het CD ($n = 10$ mice), WT HFD ($n = 10$ mice), Het HFD ($n = 10$ mice). Equal numbers of female and male mice were studied. Statistical significance was determined by Wald's test. $*p_{adj} < 0.05$. In (G), WT CD ($n = 5$ mice), Het CD ($n = 5$ mice), WT HFD ($n = 5$ mice), Het HFD ($n = 5$ mice). Statistical significance was determined by one-way ANOVA with Tukey's correction. $*p < 0.05$, $**p < 0.01$. Data are expressed as mean ± SEM. Source data are available online for this figure.

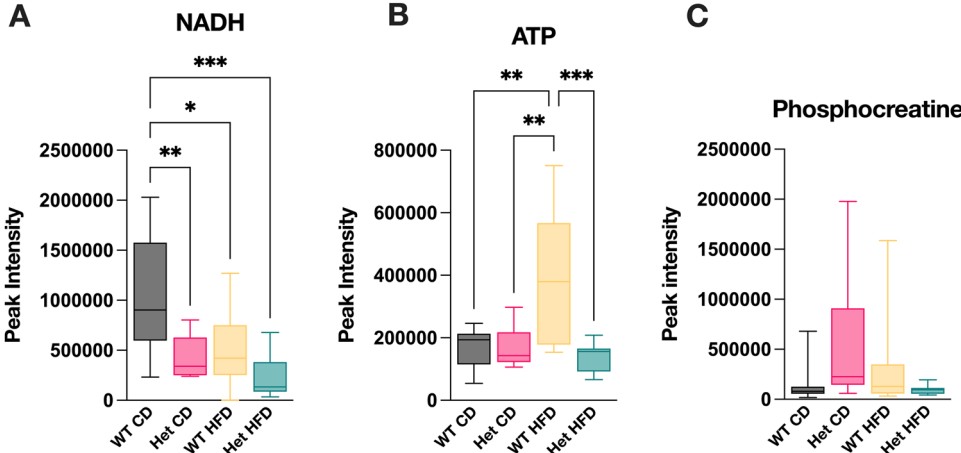

**Figure EV5. Intermediate metabolites related to energy status in the heart.**

Levels of NADH (**A**), ATP (**B**), and Phosphocreatine (**C**) in heart of Het and WT mice on CD or HFD. Data information: In (**A–C**), WT CD ($n = 10$ mice), Het CD ($n = 10$ mice), WT HFD ($n = 10$ mice), Het HFD ($n = 10$ mice). Equal numbers of female and male mice were studied. Statistical significance was determined by one-way ANOVA with Tukey's correction for multiple comparisons. *$p < 0.05$, **$p < 0.01$, ***$p < 0.0005$. Data are expressed as box plots showing minimum to maximum values, median, and upper and lower quartiles. Source data are available online for this figure.

