## [Peer Review File · EMBO Molecular Medicine]

High fat diet ameliorates mitochondrial cardiomyopathy in CHCHD10 mutant mice

Nneka Southwell, Onorina L. Manzo, Sandra Bacman, Dazhi Zhao, Nicole Sayles, Jalia Dash, Keigo Fujita, Marilena D'Aurelio, Annarita Di Lorenzo, Giovanni Manfredi, and Hibiki Kawamata

Corresponding author(s): Hibiki Kawamata (hik2004@med.cornell.edu)

Review Timeline:

Submission Date:	27th Mar 23
Editorial Decision:	24th Apr 23
Resubmission Date:	28th Dec 23
Editorial Decision:	26th Jan 24
Revision Received:	25th Mar 24
Accepted:	28th Mar 24

Editor: Zeljko Durdevic

Transaction Report:

Date: 24th Apr 23 04:52:19
Last Sent: 24th Apr 23 04:52:19
From: contact@embomolmed.org
To: hik2004@med.cornell.edu
Subject: EMM-2023-17774 Decision Letter
Message: 24th Apr 2023

Decision on your manuscript EMM-2023-17774

Dear Prof. Kawamata,

Thank you for the submission of your manuscript to EMBO Molecular Medicine. We have now received feedback from the three reviewers who agreed to evaluate your manuscript.

As you will see from their reports pasted below, all referees recognize potential interest of the study, but also raise serious and partially overlapping concerns, particularly regarding the lack of evidence to support main conclusions of the study and the lack of mechanistic insight. As clear and conclusive insight into a novel, clinically relevant observation is crucial for publication in EMBO Molecular Medicine, and together with the fact that we only accept papers that receive enthusiastic support upon initial review, I am afraid that we cannot offer to consider the manuscript further.

I am sorry that I could not bring better news this time and hope that the referee comments are helpful in your continued work in this area.

Yours sincerely,

Zeljko Durdevic

***** Reviewer's comments *****

Referee #1 (Comments on Novelty/Model System for Author):

The model system is appropriate. The data presentation could be improved throughout the figures. The findings suggest that high fat diet could be beneficial in mitochondrial cardiomyopathy. However, such claims have been reported before as acknowledged by the authors (e.g., in a YME1 cKO model of mitochondrial cardiomyopathy) and the data demonstrating a functional or survival benefit is fairly limited.

Referee #1 (Remarks for Author):

Evaluation summary

In their manuscript titled "High fat diet ameliorates mitochondrial cardiomyopathy in

CHCHD10 mutant mice," Southwell et al exposed CHCHD10S55L mice to high fat diet (HFD) from life in utero throughout adulthood. Through metabolomic, transcriptomic, and lipidomic profiling of the murine hearts, they found that wildtype and mutant mice responded to HFD differently. HFD increased fatty acid utilization and reduced CHCHD10 aggregation, cardiac stress markers, and fibrosis in CHCHD10S55L heart. When started in adulthood, HFD protected CHCHD10S55L female mice from accelerated cardiomyopathy associated with pregnancy and extended their survival. They additionally showed a benefit in females (but not males) in a frataxin model of cardiomyopathy. Thus, high fat diet appears to have a similar protective effect as was shown earlier by a different group for a model of mitochondrial cardiomyopathy due to conditional heart KO of YME1. Although this finding is not entirely novel, the model chosen by the authors in the present manuscript is perhaps more relevant to human disease, however, as it explicitly models a mutation observed in patients, which has been knocked into the germline (as opposed to the conditional heart knockout of the earlier model.) The authors additionally show a modest survival benefit (in females but not males) in a model of perhaps the most common mitochondrial cardiomyopathy (due to mutations frataxin). Additionally, the authors provide detailed metabolomic and cage metabolic data to show that lipid metabolism can be increased in the setting of mitochondrial cardiomyopathy. Overall, the results are interesting and potentially important as they suggest that a high fat diet may be beneficial in mitochondrial cardiomyopathy. However, it feels preliminary. The data showing a benefit is limited to female mice that have undergone multiple pregnancies, which leaves unclear how generalizable the findings might be. Similarly, the mechanism by which high fat diet rescues the phenotype is unclear. Additionally, the data presentation is also at times unclear and would benefit from a model demonstrating how the authors speculate a high fat diet may be beneficial in this model.

Major concerns

1. The manuscript would be greatly improved by assessing the effect of high fat diet on motor function, heart function, and/or survival in both sexes of the CHCHD10 S59L model. At present the main claims for benefit of high fat diet in mitochondrial cardiomyopathy are quite limited, as it has only been tested in females that have had multiple pregnancies. It also not clear if the benefit is specifically during pregnancy or if the benefit is more general in females and the pregnancy acts as an additional physiological stress. If the benefit is limited to females with multiple pregnancies what is the mechanism?
2. The observation that CHCHD10 total protein and protein aggregates are decreased by HFD is interesting and potentially important as it has bearing on how generalizable the benefit of HFD may be for mitochondrial cardiomyopathy. If the mechanism is reduction of misfolded CHCHD10, the benefit may be more limited to specific forms mitochondrial cardiomyopathy driven by protein misfolding. If the benefit is downstream than it may be more widely generalizable to other forms of cardiomyopathy. The authors speculate that autophagy is enhanced by HFD and may improve the protein aggregation. The authors show some benefit of HFD also in the Frataxin model. Are autophagy, p62, and protein insolubility similarly changed in the Frataxin model? If not, perhaps that supports the idea that at least some of the benefit is downstream of the effects on CHCHD10 aggregation.

Minor concerns

1. Page 10, line 30, "...while Het HFD hearts were significantly increased in Het HFD hearts". Please correct the typo.
2. Recommend providing a title for each figure.

3. Figure 1A. Authors state in the figure legend that the threshold is set at FC 2, but in the figure, the line is at Log2 fold change =2, which is a fold change of 4.
4. Figure 2A and 5A. Authors state in the figure legend that the threshold is set at FC 1, but in the figure, the line is at Log2 fold change =1, which is a fold change of 2.
5. Figure 2I, the heatmap has some duplicated names for metabolites (e.g., DL-O-Phosphoserine and A-Ketoglutaric acid oxime). Perhaps this represents distinct features for the same metabolite? If so, it may be better to include only the feature felt to best reflect levels of the metabolite.
6. Figure 2C and F, please check the labelling on the x-axis. The first column is labelled as WT CD which would imply that WT CD is being compared to WT CD. If that were the case all the log2 FC values should be 0.
7. Figure 6B and 6E-K. These charts do not have legends. It is unclear whether the labels are all the same as in 6A.
8. Figure 7E and 7F. Recommend labeling 7E and 7F in the same style as 7D (labeling the groups below each column).
9. Figure 7G and 7H. Recommend providing quantification.
10. Figure 8A. Recommend spelling out CHCHD10 instead of using the abbreviation.
11. Figure 8A, 8G, 8I, and 8K. Please provide clearer examples of your Western blot. Recommend blotting and quantifying for LC3-1 and LC3-2. Recommend comparing p62 punctae by immunofluorescence. Recommend checking protein aggregation also at a later timepoint (such as at P125) to see if the difference in proteotoxic stress persists. Recommend showing phospho-eIF2alpha/eIF2alpha levels and other markers of mtISR (e.g., MTHFD2).
12. Supplementary Figure 3 A-D. Recommend using the same color and the same order for each experimental group in all the charts throughout the paper. Het CD, Het HFD, WT CD, and WT HFD are labelled differently in 3A and 3B than in 3C and 3D. This is also different from the labeling in Supplementary Figure 4.
13. Supplementary Figure 4 A-D. Recommend using the same color and the same order for each experimental group in all the charts. WT CD and Het HFD are labelled differently in Supplementary Figure 4A than in 4B-D.
14. Supplementary Figure 4. Please provide the reason why only female mice were used in this experiment.
15. Supplementary Figure 6 B. Recommend increasing number of animals used for the FXN survival study.
16. Supplementary Figure 6 C. Please explain your graph. The label of your y-axis (mean age) and your caption (median age) do not make sense for a column chart with n= 4 mice per group.

Referee #2 (Comments on Novelty/Model System for Author):

While this study provides a large amount of metabolomic, transcriptomic and proteomic data, it provides limited new insights into how a S55L mutation of CHCHD10 results in mitochondrial cardiomyopathy. While it is proposed that administering a HFD can switch the heart from glycolysis to fatty acid oxidation, there is no actual data showing that this metabolic switch actually occurs. This conclusion is based primarily on indirect metabolic and transcriptional changes in the HET mice, and not direct metabolic flux measurements. Furthermore, the study lacks any functional data to show that the HFD is actually improving cardiac function in the HET mice. The conclusion of cardiac functional benefits is limited to measurement of cardiomyopathy markers. These issues

and others concerns are highlighted in the "Comments to Authors".

Referee #2 (Remarks for Author):

General Comments:

Previous studies have shown that CHCHD10 knock-in mice harboring a heterozygous S55L mutation develop a fatal mitochondrial cardiomyopathy. This study exposed heterozygous S55L (HET) mice to a chronic high fat diet (HFD) to decrease insulin sensitivity and glucose uptake and enhance fatty acid utilization in the heart.

Metabolomic and transcriptomic profiles demonstrated that HFD increased fatty acid utilization in the heart and ameliorated cardiomyopathy markers. It is concluded that metabolic alterations can be effectively targeted for therapeutic intervention in mitochondrial cardiomyopathies associated with proteotoxic stress.

While this study provides a large amount of metabolomic, transcriptomic and proteomic data, it provides limited new insights into how a S55L mutation of CHCHD10 results in mitochondrial cardiomyopathy. While it is proposed that administering a HFD can switch the heart from glycolysis to fatty acid oxidation, there is no actual data showing that this metabolic switch actually occurs. This conclusion is based primarily on metabolic and transcriptional changes in the HET mice, and not direct metabolic flux measurements. Furthermore, the study lacks any functional data to show that the HFD is actually improving cardiac function in the HET mice. The conclusion of cardiac functional benefits is limited to measurement of cardiomyopathy markers.

Specific Comments;

1) The assumption is made that HET mice have a switch in fatty acid oxidation to glycolysis as a source of ATP production. However, it could just as easily be due to a decrease in overall mitochondrial oxidative metabolism, with a parallel increase in glycolysis. There is no evidence as to what is happening to cardiac glucose oxidation in these mice. Glycolysis is not a major source of ATP in the heart, and is not able to compensate for major changes in mitochondrial oxidative phosphorylation. As a result, the HET mice hearts may be energy starved. Unfortunately, there is no data on energy status of these hearts.

2) The HET mice fed a HFD had major changes overall RER, body weight, physical activity and glucose tolerance. Any of these parameters could be altering cardiac energy metabolism independent of direct effects of a HFD in HET mice in the heart.

3) The authors state that: "However, in pathologic conditions, such as pressure overload, the diseased heart relies heavily on glucose uptake and glycolysis for energy production". This is a poor description of the literature, and ignores the fact that glucose oxidation is actually decreased in the failing heart.

4) The HFD did not seem to have the predicted response in the WT mice. They did not develop glucose intolerance, and cardiac fatty acid oxidative enzymes were not up-regulated. How do the authors explain this?

5) It is proposed that a HFD decreases the accumulation of aggregated CHCHD10 in the S55L heart. However, it also decreased CHCHD10 expression. Are the decreases in CHCHD10 aggregation just occurring secondary to less CHCHD10 expression?

6) The data on HFD effects on pregnancy and spinal cord metabolism do not really fit in this paper.

7) It is stated that individual ablation of CHCHD10 or CHCHD2 impairs mitochondrial function, while knock out of both proteins causes mitochondrial alterations, and that CHCHD10-related diseases are caused by a pathogenic gain of toxic function of the mutant protein. I am not sure how this ablation data supports a pathogenic gain of function role for CHCHD10.

8) Figure 3B: the authors did not see an increase in fatty acid oxidative enzymes in WT HFD, as would be expected.

- 9) Figure 3C did not examine pyruvate dehydrogenase, which would give some insights as to what happens to the oxidation of glucose.
- 10) Figure 6G is confusing. Why did the HET mice on a normal diet run the most?
- 11) Figure 7: The authors did not get the predicted GTT in WT mice on a HFD. Why?

Referee #3 (Comments on Novelty/Model System for Author):

Very interesting work performed on a well-characterized mouse model with robust multiomics studies. The medical interest is obvious since it consists in analyzing the potential and the mechanism of action of a ketogenic diet in patients with mitochondrial cardiomyopathy. In terms of review, the positive impact of the ketogenic diet on the clinico-biological level needs to be reinforced.

Referee #3 (Remarks for Author):

The paper from Southwell and colleagues reports an interesting preclinical study showing that HFD may be relevant in CHCHD10-related mitochondrial cardiomyopathy. This work is based on robust multiomics studies. The main remarks concern the clinico-biological validation of the positive effects observed in the mouse model. A ketogenic diet is used in some patients with a mitochondrial disease, mainly with refractory seizures. Positive effects on mitochondrial cardiomyopathy have also been described (Deberles et al, 2020). HFD is thought to exert its positive effects via stimulation of mitochondrial biogenesis and decrease of oxidative stress. However, it is particularly important to understand precisely the pathways involved in order to avoid adverse events and to clarify indications in clinical practice.

Major comments

- Omics profiles show that HFD induces FA utilization in the heart of KI mice and the authors report a cardioprotective effect via decreased cardiac stress markers. In their description of the KI mouse model (Anderson et al, 2019), they found objective signs of cardiomyopathy and heart failure with a systolic dysfunction by echocardiography. What are the effects of HFD on ultrasound settings?
- In the same way, the authors and others described that most RC complexes were down regulated in terms of protein levels and activity in the heart of KI mice. Mitochondrial DNA depletion also found in the heart tissues of KI mouse is at least partly mtISR-dependent. What is the effect of HFD on these markers of OXPHOS impairment and on mtDNA copy number? These are important data that cannot be studied in patients and that a preclinical trial can monitor.
- CHCHD10 forms aggregates with CHCHD2. Do the 2 proteins show the same profiles by WB and filter trap analysis after HFD? The results showing that HFD decreased accumulation of CHCHD10 aggregates is very interesting. They suggest that a metabolic intervention could be a therapeutic option in protein aggregation diseases. In Sayles et al, the authors found CHCHD10 and CHCHD2 aggregates, as early as 75 days, with an activation of mtISR at the same moment. Here, they show that HFD decreases mtISR marker levels. Do the authors have a testable hypothesis linking aggregates/mitochondrial stress/HFD effect?
- It has been reported by different teams that the expression of the CHCHD10 S59L variant is responsible for OMA1 activation. The authors should show the expression

levels of OMA1 under their experimental conditions. OMA1 activation leads to an increase of OPA1 short forms and the authors suggest that HFD has no effect on OPA1 processing in their model. Since OMA1 has been found to signal mitochondrial stress to the nucleus through the ISR, it would be interesting to determine if HFD has an effect on the level of activation of this protease.

- Shamma et al (2022) generated a novel KI mouse model expressing the Chchd10 G58R variant, which developed a cardiomyopathy. They showed that mtISR, via OMA1 activation, has a protective effect in their model. Could the authors comment on these results in relation to their data showing that the decrease of mtISR marker levels would be associated with a cardioprotective effect?

- The hypothesis that HFD decreases aggregate levels due to increased autophagy requires more robust data.

Minor comments

- The nomenclature must be revised: human gene in upper case, italic; murine gene in lower case, italic...

- In the introduction section, the authors state that CHCHD10 associated with prohibitin. In the cited article (Genin et al, 2022), the interaction is between CHCHD10 and SLP2, both participating in the prohibitin complex stability.

- KO of both Chchd2 and Chchd10 causes "mitochondrial alterations" should be clarified by mentioning that the animals partially phenocopied mutant Chchd10 KI mice with the development of cardiomyopathy and activation of the mtISR response in affected tissues.

As a service to authors, EMBO provides authors with the possibility to transfer a manuscript that one journal cannot offer to publish to another EMBO publication. The full manuscript and if applicable, reviewers reports are automatically sent to the receiving journal to allow for fast handling and a prompt decision on your manuscript. For more details of this service, and to transfer your manuscript to another EMBO title please click on *Link Unavailable*

26th Jan 2024

Dear Prof. Kawamata,

Thank you for the resubmission of your manuscript to EMBO Molecular. We have now received feedback from the three reviewers who agreed to evaluate your manuscript. As you will see from their reports pasted below all three referees support publication of your manuscript. Therefore, I am pleased to inform you that we will be able to accept your manuscript pending the following final amendments:

- 1) Please address all referee #3 concerns.
- 2) Figures: Please upload individual, high-resolution file for each main and EV figure. 5 Suppl. Figures should be made Figure EV 1-5 and the legends should stay in the manuscript. Rest of Suppl. Figures should be placed in an "Appendix" - combined in a single PDF file, with a ToC, legend removed from manuscript text and added underneath the figure, and the figure renamed "Appendix Figure S1". Please also update all figure callouts in the main text. For more information on figure presentation please check "Author Guidelines". <https://www.embopress.org/page/journal/17574684/authorguide#datapresentationformat>
- 3) Author checklist: Please submit a complete checklist. <https://www.embopress.org/pb-assets/embosite/EMBO%20Press%20Author%20Checklist-1642513524327.xlsx>
- 4) In the main manuscript file, please do the following:
 - Please address all comments suggested by our data editors listed below:
 - o Figure legends:
 1. Please note that a separate 'Data Information' section is required in the legends of figures 1a-c, e-f, h; 2a-i; 3a-d, f; 6b-d, f-h; 8b, d-e, g, i, k, m, o, q; supplementary figures 2a, c; 3a-d; 4a-d; 5a-g; 6a-f; 7a-c.
 2. Please note that the figure 2g does not contain asterisk, please rectify the statistical test related information in the legend. Kindly look into this.
 3. Please note that the legend for figure 7a is provided as 7b, similarly the legend for figure 7b is provided as 7a. This needs to be rectified. Further, we have evaluated the figures as per the labelling provided in the legend.
 4. Please indicate the statistical test used for data analysis in the legends of figures 4a-g; 5a-c, e-h; 6b-c, f; 7b, e; supplementary figures 5a-d, f-g; 6a-f.
 5. Please note that the box plots need to be defined in terms of centre, bounds of box and percentile in the legends of figures 6d, g-h; 7a-c.
 6. Please note that information related to n is missing in the legends of figures 4a-d; 5a-c; 6d, g-h; supplementary figures 1a-l; 5a-c.
 7. Although 'n' is provided, please describe the nature of entity for 'n' in the legends of figures 2d-i; 8g, i, k, m, o, q.
 - Rename "Methods" to "Materials and Methods".
 - In M&M, provide the antibody dilutions that were used for each antibody.
 - Remove abbreviation list.
 - Add "Disclosure Statement & Competing Interests". We updated our journal's competing interests policy in January 2022 and request authors to consider both actual and perceived competing interests. Please review the policy <https://www.embopress.org/competing-interests> and update your competing interests if necessary.
 - Author contributions: Please remove it from the manuscript and specify author contributions in our submission system. CRediT has replaced the traditional author contributions section because it offers a systematic machine-readable author contributions format that allows for more effective research assessment. You are encouraged to use the free text boxes beneath each contributing author's name to add specific details on the author's contribution. More information is available in our guide to authors: <https://www.embopress.org/page/journal/17574684/authorguide#authorshipguidelines>
 - Add data availability statement. This paragraph should contain information about deposited data produced in this study. Please be aware that large-scale data (e.g. 3'-RNA seq, metabolomics etc.) need to be deposited in a public repository and made freely available. Please deposit the raw data from all large-scale experiments in an appropriate public repository and use the following format to report the accession number of your data:

[data type]: [full name of the resource] [accession number/identifier] ([doi or URL or identifiers.org/DATABASE:ACCESSION])

Please check "Author Guidelines" for more information.

<https://www.embopress.org/page/journal/17574684/authorguide#availabilityofpublishedmaterial>

- 5) Funding: Please merge it with "Acknowledgement. Also make sure that information about all sources of funding are complete in both our submission system and in the manuscript. Currently, NextGenALS 2021-01 to HK and GM is missing in our submission system.
- 6) The Paper Explained: Please provide "The Paper Explained" and add it to the main manuscript text. Please check "Author Guidelines" for more information. <https://www.embopress.org/page/journal/17574684/authorguide#researcharticleguide>
- 7) Synopsis: Every published paper now includes a 'Synopsis' to further enhance discoverability. Synopses are displayed on the journal webpage and are freely accessible to all readers. They include separate synopsis image and synopsis text.

- Synopsis image: Please provide a striking image or visual abstract as a high-resolution jpeg file 550 px-wide x (250-400)-px high to illustrate your article.

- Synopsis text: Please provide a short standfirst (maximum of 300 characters, including space) as well as 2-5 one sentence bullet points that summarise the paper as a .doc file. Please write the bullet points to summarise the key NEW findings. They should be designed to be complementary to the abstract - i.e. not repeat the same text. We encourage inclusion of key acronyms and quantitative information (maximum of 30 words / bullet point). Please use the passive voice.

8) For more information: This space should be used to list relevant web links for further consultation by our readers. Could you identify some relevant ones and provide such information as well? Some examples are patient associations, relevant databases, OMIM/proteins/genes links, author's websites, etc...

9) As part of the EMBO Publications transparent editorial process initiative (see our Editorial at <http://embomolmed.embopress.org/content/2/9/329>), EMBO Molecular Medicine will publish online a Review Process File (RPF) to accompany accepted manuscripts. This file will be published in conjunction with your paper and will include the anonymous referee reports, your point-by-point response and all pertinent correspondence relating to the manuscript. Let us know whether you agree with the publication of the RPF and as here, if you want to remove or not any figures from it prior to publication. Please note that the Authors checklist will be published at the end of the RPF.

10) Please provide a point-by-point letter INCLUDING my comments as well as the reviewer's reports and your detailed responses (as Word file).

I look forward to reading a new revised version of your manuscript as soon as possible.

Yours sincerely,

Zeljko Durdevic

*** Instructions to submit your revised manuscript ***

1) a .docx formatted version of the manuscript text (including Figure legends and tables)

2) Separate figure files*

3) supplemental information as Expanded View and/or Appendix. Please carefully check the authors guidelines for formatting Expanded view and Appendix figures and tables at <https://www.embopress.org/page/journal/17574684/authorguide#expandedview>

4) a letter INCLUDING the reviewer's reports and your detailed responses to their comments (as Word file).

5) The paper explained: EMBO Molecular Medicine articles are accompanied by a summary of the articles to emphasize the

major findings in the paper and their medical implications for the non-specialist reader. Please provide a draft summary of your article highlighting

This may be edited to ensure that readers understand the significance and context of the research.

Please refer to any of our published articles for an example.

6) For more information: There is space at the end of each article to list relevant web links for further consultation by our readers. Could you identify some relevant ones and provide such information as well? Some examples are patient associations, relevant databases, OMIM/proteins/genes links, author's websites, etc...

7) Author contributions: the contribution of every author must be detailed in a separate section.

8) EMBO Molecular Medicine now requires a complete author checklist

(<https://www.embopress.org/page/journal/17574684/authorguide>) to be submitted with all revised manuscripts. Please use the checklist as guideline for the sort of information we need WITHIN the manuscript. The checklist should only be filled with page numbers where the information can be found. This is particularly important for animal reporting, antibody dilutions (missing) and exact values and n that should be indicated instead of a range.

9) Every published paper now includes a 'Synopsis' to further enhance discoverability. Synopses are displayed on the journal webpage and are freely accessible to all readers. They include a short stand first (maximum of 300 characters, including space) as well as 2-5 one sentence bullet points that summarise the paper. Please write the bullet points to summarise the key NEW findings. They should be designed to be complementary to the abstract - i.e. not repeat the same text. We encourage inclusion of key acronyms and quantitative information (maximum of 30 words / bullet point). Please use the passive voice. Please attach these in a separate file or send them by email, we will incorporate them accordingly.

You are also welcome to suggest a striking image or visual abstract to illustrate your article. If you do please provide a jpeg file 550 px-wide x 300-800px high.

10) A Conflict of Interest statement should be provided in the main text

11) Please note that we now mandate that all corresponding authors list an ORCID digital identifier. This takes <90 seconds to complete. We encourage all authors to supply an ORCID identifier, which will be linked to their name for unambiguous name identification.

Currently, our records indicate that the ORCID for your account is 0000-0003-0020-6933.

Link Not Available

Photos 400-800 DPI

*Additional important information regarding figures and illustrations can be found at

<https://bit.ly/EMBOPressFigurePreparationGuideline>. See also figure legend preparation guidelines:

<https://www.embopress.org/page/journal/17574684/authorguide#figureformat>

***** Reviewer's comments *****

Referee #1 (Remarks for Author):

The authors have addressed all of the concerns I raised in my initial review of the manuscript. This includes adding new experimental data. I therefore feel that the paper is now acceptable for publication.

Referee #2 (Comments on Novelty/Model System for Author):

The revision is much approved and addresses my major concerns. Thank you for this excellent work.

Referee #2 (Remarks for Author):

The revision is much approved and addresses my major concerns. Thank you for this excellent work.

Referee #3 (Comments on Novelty/Model System for Author):

In the manuscript titled « High fat diet ameliorates mitochondrial cardiomyopathy in CHCHD10 mutant mice», Southwell et al. report an interesting study showing that chronic HFD may be important in CHCHD10-related mitochondrial cardiomyopathy. They show that HFD improves cardiac dysfunction in Chchd10S55L/+ mice and significantly prolongs survival of Het mice. This work is based on transcriptomics and metabolomics analyses in mice treated with HFD throughout their life in utero by feeding pregnant dams with HFD and continuing the diet after weaning. The results show a rearrangement in metabolism that enhances FAO and decreases cardiomyopathy markers. To better understand the molecular mechanism underlying the cardioprotective effects of HFD, the authors show that HFD decreases CHCHD10 protein levels and the aggregation of CHCHD10 and CHCHD2 in Het hearts.

Referee #3 (Remarks for Author):

In the manuscript titled « High fat diet ameliorates mitochondrial cardiomyopathy in CHCHD10 mutant mice», Southwell et al. report an interesting study showing that chronic HFD may be important in CHCHD10-related mitochondrial cardiomyopathy. They show that HFD improves cardiac dysfunction in Chchd10S55L/+ mice and significantly prolongs survival of Het mice. This work is based on transcriptomics and metabolomics analyses in mice treated with HFD throughout their life in utero by feeding pregnant dams with HFD and continuing the diet after weaning. The results show a rearrangement in metabolism that enhances FAO and decreases cardiomyopathy markers. To better understand the molecular mechanism underlying the cardioprotective effects of HFD, the authors show that HFD decreases CHCHD10 protein levels and the aggregation of CHCHD10 and CHCHD2 in Het hearts.

Remarks to the authors :

- Multi-omics experiments indicate that HFD reduces cardiac stress and cardiomyopathy markers. It would be interesting to correlate these transcriptomic results at the protein level.
- Mitochondrial DNA depletion, which was also found in the heart tissue of Het mice, is at least partially dependent on mtISR. What is the impact of HFD on these parameters of OXPHOS impairment and mtDNA copy number ?
- The authors investigated the mitochondrial levels of two key mitophagy players, PINK1 and Parkin. They conclude that the levels of these two proteins were elevated in Het in CD and significantly decreased by HFD. Based on the WB shown in Fig.8L, it is difficult to conclude that PINK1 is increased in Het in CD. Furthermore, the authors conclude that HFD increases mitophagy in Het hearts, but at the same time show that the expression of Pink1, Parkin, p62 and LC3B decreases under HFD compared to Het in CD. The role of mitophagy in the response to HFD is not clear and needs further investigation and/or clarification.

We thank the reviewers and the Editor for their positive assessment of our revised manuscript.

Response to reviewer 3:

1) Multi-omics experiments indicate that HFD reduces cardiac stress and cardiomyopathy markers. It would be interesting to correlate these transcriptomic results at the protein level.

We performed Western blot analyses of two proteins associated with cardiac stress, PMCA (*Atp2b4*) and MYH6. The results shown in new Appendix figure S6 indicate that HFD decreases the levels of these proteins.

2) Mitochondrial DNA depletion, which was also found in the heart tissue of Het mice, is at least partially dependent on mtISR. What is the impact of HFD on these parameters of OXPHOS impairment and mtDNA copy number.

We thank the reviewer for this suggestion. We evaluated mtDNA levels by Droplet PCR in the heart of Het mice at 75 days treated with control or HFD and found that Het mice had approximately 30% mtDNA depletion and that the diet had no effect on mtDNA depletion. Therefore, we concluded that the beneficial effects of HFD are not dependent on mtDNA recovery. The mtDNA quantifications have been performed by Dr. Sandra Bacman at University of Miami, who is now one of the co-authors of the manuscript.

3) The authors investigated the mitochondrial levels of two key mitophagy players, PINK1 and Parkin. They conclude that the levels of these two proteins were elevated in Het in CD and significantly decreased by HFD. Based on the WB shown in Fig.8L, it is difficult to conclude that PINK1 is increased in Het in CD. Furthermore, the authors conclude that HFD increases mitophagy in Het hearts, but at the same time show that the expression of Pink1, Parkin, p62 and LC3B decreases under HFD compared to Het in CD. The role of mitophagy in the response to HFD is not clear and needs further investigation and/or clarification.

We apologize for the lack of clarity on this point. We have now added a new exposure of the Pink1 Western blot to show that the protein levels associated with Het heart mitochondria is higher than in WT. Furthermore, we clarify in the result and discussion sections our interpretation of the data of figure 8. We propose that in Het hearts mitophagy is initiated due to mitochondrial damage, but it fails to progress to the stage of organellar degradation, whereas in HFD the turnover of the mitochondria tagged for mitophagy is more efficient. We also acknowledge that more work will be needed in the future to further investigate the mechanisms of mitophagy impairment in Het hearts.

Response to editorial comments:

1) Figures: Please upload individual, high-resolution file for each main and EV figure. 5 Suppl. Figures should be made Figure EV 1-5 and the legends should stay in the manuscript. Rest of Suppl. Figures should be placed in an "Appendix" - combined in a single PDF file, with a ToC, legend removed from manuscript text and added underneath the figure, and the figure renamed "Appendix Figure S1". Please also update all figure callouts in the main text.

We now have 5 EV figures and 4 appendix figures. The callouts and legends have been organized as per journal instructions.

2) Author checklist: Please submit a complete checklist.

The checklist has been submitted.

3) In the main manuscript file, please do the following:

Please address all comments suggested by our data editors listed below:

Figure legends:

- 1. Please note that a separate 'Data Information' section is required in the legends of figures 1a-c, e-f, h; 2a-i; 3a-d, f; 6b-d, f-h; 8b, d-e, g, i, k, m, o, q; supplementary figures 2a, c; 3a-d; 4a-d; 5a-g; 6a-f; 7a-c.*
- 2. Please note that the figure 2g does not contain asterisk, please rectify the statistical test related information in the legend. Kindly look into this.*
- 3. Please note that the legend for figure 7a is provided as 7b, similarly the legend for figure 7b is provided as 7a. This needs to be rectified. Further, we have evaluated the figures as per the labelling provided in the legend.*
- 4. Please indicate the statistical test used for data analysis in the legends of figures 4a-g; 5a-c, e-h; 6b-c, f; 7b, e; supplementary figures 5a-d, f-g; 6a-f.*
- 5. Please note that the box plots need to be defined in terms of centre, bounds of box and percentile in the legends of figures 6d, g-h; 7a-c.*
- 6. Please note that information related to n is missing in the legends of figures 4a-d; 5a-c; 6d, g-h; supplementary figures 1a-l; 5a-c.*
- 7. Although 'n' is provided, please describe the nature of entity for 'n' in the legends of figures 2d-i; 8g, i, k, m, o, q.*

The changes have been made.

4) *Rename "Methods" to " Materials and Methods".*

Changed

5) *In M&M, provide the antibody dilutions that were used for each antibody.*

Provided.

6) *Remove abbreviation list.*

Removed.

7) *Add "Disclosure Statement & Competing Interests".*

Disclosure added.

8) *Author contributions: Please remove it from the manuscript and specify author contributions in our submission system.*

Removed from main text and submitted on journal website.

9) *Add data availability statement. This paragraph should contain information about deposited data produced in this study. Please be aware that large-scale data (e.g. 3'-RNA seq, metabolomics etc.) need to be deposited in a public repository and made freely available. Please deposit the raw data from all large-scale experiments in an appropriate public repository.*

The statement regarding RNA and metabolites has been added and the information about the deposited data provided with relative numbers.

10) *Funding: Please merge it with "Acknowledgement. Also make sure that information about all sources of funding is complete in both our submission system and in the manuscript. Currently,*

NextGenALS 2021-01 to HK and GM is missing in our submission system.

Funding has been merged with acknowledgments and the missing finding sources added.

11) *The Paper Explained: Please provide "The Paper Explained" and add it to the main manuscript text.*

The paper explained has been added.

12) *Synopsis: Every published paper now includes a 'Synopsis' to further enhance discoverability. Synopses are displayed on the journal webpage and are freely accessible to all readers.*

The synopsis image and text are provided as a separate document.

13) *For more information: This space should be used to list relevant web links for further consultation by our readers.*

Not applicable.

14) *As part of the EMBO Publications transparent editorial process initiative (see our Editorial at <http://embomolmed.embopress.org/content/2/9/329>), EMBO Molecular Medicine will publish online a Review Process File (RPF) to accompany accepted manuscripts. This file will be published in conjunction with your paper and will include the anonymous referee reports, your point-by-point response and all pertinent correspondence relating to the manuscript. Let us know whether you agree with the publication of the RPF and as here, if you want to remove or not any figures from it prior to publication. Please note that the Authors checklist will be published at the end of the RPF.*

We agree to publishing an RPF including the revised manuscript and figures (first resubmission) and correspondence. However, we don't think that the first submission of the manuscript would make sense to publish in the RPF because text and figures have been changed very extensively with a completely different order of topics and removal of all the CNS studies.

28th Mar 2024

Dear Prof. Kawamata,

We are pleased to inform you that your manuscript is accepted for publication and is now being sent to our publisher to be included in the next available issue of EMBO Molecular Medicine.

Yours sincerely,
